# Explicit Exploration for High-Welfare Equilibria in Game-Theoretic Multiagent Reinforcement Learning

Austin A. Nguyen [1]   Anri Gu [1]   Michael P. Wellman [1]

## Abstract

Iterative extension of empirical game models through deep reinforcement learning (RL) has proved an effective approach for finding equilibria in complex games. When multiple equilibria exist, we may have preferences among solutions. We address this equilibrium selection issue in the context of Policy Space Response Oracles (PSRO), a flexible game-solving framework based on deep RL, by skewing strategy generation towards higher-welfare solutions. At each iteration, we create an *exploration policy* that imitates high welfare-yielding behavior and train a response to the current solution, regularized to be similar to the exploration policy. With no additional simulation expense, our approach, named $\text{Ex}^2\text{PSRO}$, tends to find higher welfare equilibria than vanilla PSRO in two benchmarks: a sequential bargaining game and a social dilemma game. Further experiments demonstrate $\text{Ex}^2\text{PSRO}$'s composability with other PSRO variants and illuminate the relationship between exploration policy choice and algorithmic performance.

## 1. Introduction

Multiagent reinforcement learning (MARL) research addresses two core challenges: *scalability* and *non-stationarity*. Scalability refers to how the relevant search space for training grows exponentially with the number of players and strategies. Non-stationarity describes how agents' learning contexts change as their observations and rewards depend on the evolving strategies of other agents. If the goal is to specify policies for distributed agents under unified control, these challenges can be addressed under the paradigm of **centralized training decentralized execution** (Rashid et al., 2020; Lowe et al., 2017).

When the agents are independently controlled, it makes sense to frame the problem in terms of solving a game. In a game formulation, non-stationarity can be understood as arising from strategic interactions among agent decisions. Our work builds on a standard framework in this genre called **Policy Space Response Oracles** (PSRO) (Lanctot et al., 2017). PSRO uses a form of population-based training, where the set of available policies (strategies) is iteratively extended by deriving best responses to the evolving game model. PSRO can also be viewed as a form of **empirical game-theoretic analysis** (EGTA) (Wellman et al., 2025), as the game model is estimated through agent-based simulation. PSRO addresses scalability and non-stationarity by focusing on an enumerated set of strategies from a large space and training one agent at a time in a context where other agents' strategies are fixed.

The efficacy of such iterative methods depends on the ability to identify strategies that cover the relevant game features and contribute to game solutions. Alternative approaches to **strategy exploration** (Jordan et al., 2010) generate different sequences of game models and thus determine how quickly a solution is found (Wang et al., 2022). The method of strategy exploration can also shape *which* solution is found, that is, influence **equilibrium selection** for games with multiple strategic equilibria (Wang & Wellman, 2024).

This work addresses equilibrium selection by steering strategy exploration toward solutions with desirable characteristics. Specifically, we extend PSRO by leveraging the simulation data already generated by the algorithm to identify policy elements that seem promising for high-welfare solutions. Using behavior cloning, we train an **exploration policy** that reflects the promising behavior found in simulation traces. We then modify the process of training a best-response strategy, by adding a regularization term that biases the output to be similar to the exploration policy. This *ex*plicit form of controlled *ex*ploration in policy space gives the algorithm its name: $\text{Ex}^2\text{PSRO}$.

The primary contribution of this work is a method based on policy-level regularization for strategy exploration in service

---

[1]Computer Science & Engineering, University of Michigan, Ann Arbor, MI, United States. Correspondence to: Austin A. Nguyen <ngaustin@umich.edu>, Anri Gu <anrigu@umich.edu>.

*Proceedings of the $42^{nd}$ International Conference on Machine Learning*, Vancouver, Canada. PMLR 267, 2025.

of equilibrium selection. Through experiments on two game classes for which equilibrium selection is paramount—a social dilemma game and a bargaining game—we find that Ex$^2$PSRO tends to produce higher welfare solutions than the PSRO baseline. Further experiments combining Ex$^2$PSRO with existing high-welfare equilibrium selection approaches demonstrate its modularity and ability to improve welfare on other PSRO variants.

## 2. Terminology and Notation

A *symmetric game* $\Gamma$ is represented in *normal form* by a tuple $(\Pi, U, n)$, where $n$ is the number of players, $\Pi$ their set of available strategies, and *payoff function* $U : \Pi^n \to \mathbb{R}^n$ maps joint strategies $\pi = (\pi_1, \ldots, \pi_n)$ to vectors of payoffs. Symmetry dictates that $U$ be invariant to permutations of player position: $U(perm(\pi)) = perm(U(\pi))$.[1] Policy $\pi \in \Pi$ is called a *pure strategy*, and $\sigma \in \Delta(\Pi)$ is a *mixed strategy*. A subscript $-1$ indicates all players except one. For instance, $\pi_{-1}$ is the joint strategy of all other players, with "other" clear from context. We write the payoff for playing $\pi$ when the others play $\pi_{-1}$ as $u(\pi, \pi_{-1})$, where $u$ refers to the element of vector $U$ indicating the payoff given to the player invoking strategy $\pi$. When the others play joint mixed strategy $\sigma_{-1}$, the payoff is given by expectation:

$$u(\pi, \sigma_{-1}) = \mathbb{E}_{\pi_{-1} \sim \sigma_{-1}} u(\pi, \pi_{-1}). \tag{1}$$

Given our symmetry assumption, we often overload notation to allow a strategy (pure or mixed) to stand for a symmetric joint strategy (i.e., all play the same). The *welfare* of $\sigma$ refers to the expected summed payoffs across all players when playing strategy $\sigma$.

Likewise under symmetry, a player's *best response* to all others playing $\sigma$ is given by $BR(\sigma) = \arg\max_{\pi \in \Pi} u(\pi, \sigma_{-1})$. Mixed strategy $\sigma^*$ is a (symmetric) *Nash equilibrium* (NE) iff $\sigma^* \in BR(\sigma^*)$. The potential gains of deviating from symmetric play of $\sigma$ is called *regret*: $Regret(\sigma) = \max_{\pi' \in \Pi} u(\pi', \sigma_{-1}) - \mathbb{E}_{\pi \sim \sigma} u(\pi, \sigma_{-1})$. By definition, if $\sigma^*$ is a NE, its regret is zero. An *empirical game model* $\overline{\Gamma} = (\overline{\Pi}, \overline{U}, n)$ is a normal-form game over strategy set $\overline{\Pi}$ (a finite subset of a larger policy space $\Pi$), with payoff function $\overline{U}$ estimated through simulation.

*Reinforcement learning* (RL) models the environment as a *Markov Decision Process* (MDP). A policy $\pi_\theta$ is defined as a mapping from the MDP's observation space $\mathcal{O}$ to a probability simplex over the action space $\mathcal{A}$: $\pi_\theta : \mathcal{O} \to \Delta(\mathcal{A})$. The probability of choosing action $a_t$ given observation $o_t$ is given by $\pi_\theta(a_t \mid o_t)$. When an agent applies action $a_t$ given observation $o_t$, a reward $r_t$ is observed. Each simulation of the environment generates a set of trajectories

---

[1]We assume symmetry throughout this paper for convenience, as our experiments are performed on symmetric games. Extension to non-symmetric or role-symmetric games is discussed below.

$\tau = \{\tau^1, \ldots, \tau^n\}$ (one for each player). Each trajectory is a sequence of observations, actions, and rewards for a player $p$ over time: $\tau_j^p = \{(o_t^p, a_t^p, r_t^p)\}_{t=1}^T$.

## 3. Related Work

Policy regularization is widely used in RL to instill prior behavior during training (Cheng et al., 2019; Rudner et al., 2021; Bakhtin et al., 2022). In offline RL, regularization can encourage similarity between the learned policy and the dataset's behavior, reducing distribution shift (Wu et al., 2019; Kostrikov et al., 2021; Xu et al., 2021). In online RL, self-imitation learning reinforces behaviors that yield desired outcomes in sparse-reward contexts (Oh et al., 2018). In MARL, policy regularization has been used to promote coordination (Roy et al., 2020) or as an alternative to entropy regularization (Su & Lu, 2022). Regularized Nash Dynamics (R-NaD) (Pérolat et al., 2021) demonstrated the use of regularization as part of a game-theoretic MARL regime. Approaches like Follow the Regularized Leader (FoReL) have also utilized regularization to balance the exploitation-exploration tradeoff (Pérolat et al., 2021).

The PSRO algorithm (Lanctot et al., 2017) works by iteratively extending an empirical game model $\overline{\Gamma} = (\overline{\Pi}, \overline{U}, n)$ using RL. We initialize $\overline{\Pi}$, for example with a single random policy. PSRO extends $\overline{\Gamma}$ at each iteration through these steps: (1) update $\overline{U}$ by estimating payoffs for all *strategy profiles* (i.e., joint strategies over $\overline{\Pi}$) through simulation, (2) extract a *target profile* $\sigma^\top$ by applying a *meta-strategy solver* (MSS) to $\overline{\Gamma}$, and (3) derive a *best-response policy* $\pi^\star \in \Pi$ by applying deep RL in an environment where other players are fixed to play $\sigma^\top$ and add $\pi^\star$ to $\overline{\Pi}$.

A key idea of PSRO is abstracting the selection of best-response target, through the MSS. Different MSSs generate different response sequences and hence guide strategy exploration. Using *replicator dynamics* (RD) (Taylor & Jonker, 1978) or any other Nash solver as MSS corresponds to the *double oracle* method (McMahan et al., 2003). The original PSRO paper (Lanctot et al., 2017) noted that DO overfits to the current solution and introduced projected replicator dynamics (PRD) as an MSS that produces approximate Nash equilibrium. The PSRO literature has investigated a variety of alternative MSSs, as well as other variants designed to reduce computational cost or exploit game model structure (Bighashdel et al., 2024).

A recent study by Wang & Wellman (2024) investigated how varying the response objective can affect the behavior of PSRO, including its selection of equilibria. In particular, this work showed that optimizing responses for a combination of own and other-agent utility could steer PSRO toward higher-welfare equilibria. These objectives may incorporate broader measures (e.g. total social welfare), enabling

PSRO to discover more desirable equilibria. Related research has focused on integrating diversity measures into response objectives to ensure thorough exploration of policy space (Balduzzi et al., 2019; Liu et al., 2022; Perez-Nieves et al., 2021; Parker-Holder et al., 2020).

## 4. Explicit Exploration

Consider the simple 2-player game shown in Fig. 1. There are two symmetric equilibria: playing A and B with equal probability or playing C. The latter Pareto dominates, but starting double oracle (or PSRO) from A or B would produce the former. Even if we replaced best-response with an optimization of social welfare (or any linear combination with individual payoff), C would never be selected when the other player is fixed at a mixture of A and B.

|   | A | B | C |
|---|---|---|---|
| A | 2,2 | 2,3 | 1,–2 |
| B | 3,2 | 1,1 | 2,–1 |
| C | –2,1 | –1,2 | 5,5 |

*Figure 1.* A 2-player, symmetric normal-form game. Starting from either A or B, PSRO converges to an equilibrium mixture over A and B (regardless of MSS), which is Pareto-dominated by (C, C).

This example highlights that discovering high-welfare equilibria sometimes requires players to add strategies that are currently suboptimal (i.e., with respect to any other agents' generated strategies), but could ultimately lead to more favorable equilibria. In the absence of an all-knowing coordinator, we need some way to identify and steer strategy exploration toward potentially effective joint behavior. In the context of the PSRO framework, this suggests we may need to produce strategies that could not be found through best-response operations alone, regardless of MSS.

### 4.1. Extracting Desired Behavior

To guide exploration, we first identify behavior that could lead to higher welfare equilibria. Ex²PSRO does this by scanning simulation data for traces that seem to produce high-welfare outcomes. To that end, Ex²PSRO extracts a buffer $\mathcal{B}$ of $N$ trajectories $\{\tau_j\}_j^N$ that maximize some criterion $C(\tau_j)$. In symmetric games,[2] we employ a maximin welfare criterion:

$$C(\tau_j) = \min_p(V_j^p), \qquad (2)$$

---

[2]Symmetry entails we can evaluate player payoffs on the same scale when a given trajectory plays out differently for various players. For non-symmetric games, payoff normalization (using empirically estimated minimum across players return percentile as $C(\tau_j)$) and player-specific exploration policies would be required.

where $V_j^p = \sum_{t=1}^{|\tau_j|} r_t^p$ is player $p$'s return in trajectory $\tau_j$. This criterion maximizes welfare and disregards trajectories where welfare is high at the expense of any one player. Trajectories are collected from all simulation queries (i.e., empirical game updates, response training). Among the observed trajectories $\tau_j$, $\mathcal{B}$ selects the $N$ with the highest criterion score $C(\tau_j)$.[3]

### 4.2. Constructing Exploration Policies

We reiterate that an exploration policy should introduce elements that potentially incorporate pro-welfare properties. Since our criterion $C(\tau_j)$ expressly selects for that behavior, it is reasonable that the exploration strategy should match behavior in $\mathcal{B}$. We denote an exploration policy at Ex²PSRO's $i^{th}$ iteration by $\pi_{ex}^i$. Ex²PSRO trains $\pi_{ex}^i$ by using **behavior cloning** (BC) on $\mathcal{B}$. Specifically, we create a policy that maximizes:

$$\pi_{ex}^i(a \mid o) = \underset{\pi_\theta}{\arg\max}\, \mathbb{E}(\pi_\theta(a_k \mid o_k))\ \ (o_k, a_k) \sim \mathcal{B}, \ (3)$$

where we ignore the reward component of trajectories in $\mathcal{B}$. Note that $\pi_{ex}^i$ is not itself added to the empirical game but rather used to steer $\pi_\theta^i$ (Ex²PSRO's response policy at iteration $i$), in a particular way. This is discussed in the next section. $\pi_{ex}^i$ is estimated using a neural network with a softmax output layer and is optimized using cross-entropy loss. We defer further details to Apps. A.1 and B.

### 4.3. Policy Regularization Objective

We now demonstrate how exploration policy $\pi_{ex}^i$ is used to influence the added strategy $\pi_\theta^i$. Ex²PSRO builds on the baseline's - "vanilla" PSRO's - best-response step of the algorithm. At each iteration $i$, vanilla PSRO trains an approximate *best-response* policy $\pi_\theta^{*i}$ to optimize the objective:

$$\begin{aligned} J^i(\theta) &:= \mathbb{E}_{\pi_\theta\mid i} \sum_{t=1}^T r_t, \\ \pi_\theta^{*i} &\approx \underset{\pi_\theta}{\arg\max}\, J^i(\theta), \end{aligned} \qquad (4)$$

where $r_t$ is the player's reward at time $t$, $T$ refers to the (potentially infinite) horizon, $\theta$ represents the parameters of the policy $\pi_\theta$, and $\approx$ underscores the use of deep RL to approximate the best-response. The expectation in (4) is conditioned on $i$, reflecting the best-response target selected by PSRO's MSS at that iteration.

We insert a regularization term $R^i(\theta)$ that steers $\pi_\theta^i$ to be more similar to $\pi_{ex}^i$. More formally, let $R^i(\theta)$ be a measure of the distance of a policy parameterized by $\theta$ to the

---

[3]Stochasticity in returns is not accounted for, so chance success affects the selection.

exploration policy at iteration $i$. Specifically, we define:

$$R^i(\theta) = D_{KL}(\pi_{ex}^i \parallel \pi_\theta), \tag{5}$$

where $D_{KL}(. \parallel .)$ denotes KL divergence. $R^i(\theta)$ is estimated through sampling and assumes $\mathcal{R}$, the RL oracle's replay buffer, defines the prior over observations $o_t$:

$$D_{KL}(\pi_{ex}^i \parallel \pi_\theta) \approx$$
$$\mathbb{E}_{o_t \sim \mathcal{R}} \sum_{a \in \mathcal{A}} \pi_{ex}^i(a \mid o_t) \log\left(\frac{\pi_{ex}^i(a \mid o_t)}{\pi_\theta(a \mid o_t)}\right), \tag{6}$$

Ex$^2$PSRO encourages exploration by introducing $R^i(\theta)$ as a regularization term in the response objective and weighting $R^i(\theta)$ by $\beta_i$ at iteration $i$:

$$\pi_\theta^i \approx \arg\max_{\pi_\theta} \left(J^i(\theta) - \beta_i R^i(\theta)\right). \tag{7}$$

Intuitively, the objective (7) skews the response in the direction of the exploration policy. $\pi_{ex}^i$ is constructed to reflect high-welfare-yielding behavior, so the objective encourages the introduction of strategies that contain more pro-welfare elements. Moreover, since the trajectory buffer is shared amongst all players, they are all regularized towards behaviors that ideally coordinate well with each other. This can help address the issue posed by the example in Fig. 1. Ideally, behavior in $\mathcal{B}$ and strategy C would be similar. Then, regularization toward $\mathcal{B}$'s behavior would skew strategy exploration by encouraging the addition of high-welfare yielding policies similar to strategy C. We underscore that trajectory buffer sharing does not modify game dynamics—each player still only has access to their own information. This is discussed further in App. F.

### 4.4. Ex$^2$PSRO Algorithm

Many PSRO implementations use value-based RL methods to train approximate best responses. Our explicit exploration approach employs regularization defined in policy space. As regularizing value estimates by a policy is undefined, value-based methods are unsuitable. Accordingly, we adopt the Soft Actor-Critic (SAC) (Haarnoja et al., 2018) method to train approximate best responses, specifically its discrete implementation (Christodoulou, 2019; Zhou et al., 2022). Ex$^2$PSRO (Alg. 1) modifies SAC's actor update by augmenting its objective with the regularization term from Eq. (6). For consistency in comparison, we use SAC for both Ex$^2$PSRO and vanilla PSRO's best responses, where the latter's regularization strength $\beta_i$ is set to zero for each iteration $i$.

The first iteration of Ex$^2$PSRO is nearly identical to that of PSRO. It starts by initializing random strategies for each player, creating an *unregularized* best response $\pi_\theta^{*1}$ using

---

**Algorithm 1** Ex$^2$PSRO

**Input**: Max number of trajectories $N$, meta-strategy solver $\mathcal{M}$, Initial regularization $\beta_{init}$, Regularization decay steps $S_\beta$, PSRO iterations $T$, number of players $n$
**Output**: Strategy set $\Pi$, Profile $\sigma^T$

1: Initialize empty empirical game $\overline{\Gamma} : (\Pi, \overline{U}, n)$
2: Initialize strategy set w/ random policy $\Pi \leftarrow \{\pi_\theta^0\}$
3: Initialize $\beta_0 \leftarrow \beta_{init}$
4: Update empirical game $\overline{\Gamma}$
5: Initialize $\mathcal{B}_0$ with used trajectories
6: Initialize $\mathcal{B} \leftarrow$ N trajectories in $\mathcal{B}_0$ maximizing $C(\tau_j)$
7: **for** each PSRO iteration $i$ of to $T$ **do**
8:      Compute $\pi_{ex}^i$ using $\mathcal{B}$ and Eq. 3 if $i > 1$
9:      Compute response $\pi_\theta^i$ using $\beta_i, \pi_{ex}^i, \sigma_{-1}^i$ and Eq. 7
10:     Update strategy sets $\Pi \leftarrow \Pi \cup \pi_\theta^i$
11:     Update empirical game $\overline{\Gamma}$
12:     Store used trajectories in $\mathcal{B}_i$
13:     $\mathcal{B} \leftarrow$ Top N trajectories $\tau_j \in \{\mathcal{B}_i \cup \mathcal{B}\}$ w/ Eq. 2
14:     Update joint profile $\sigma^{i+1} \leftarrow \mathcal{M}(\overline{\Gamma})$
15:     Anneal $\beta_i \leftarrow \max(0, \frac{\beta_{init}(S_\beta - i)}{S_\beta})$
16: **end for**
17: **return** $\Pi, \sigma^T$

---

deep reinforcement learning, updating the empirical game $\overline{\Gamma}$, and updating meta-strategies $\mathcal{M}(\overline{\Gamma})$. Ex$^2$PSRO creates an unregularized best response $\pi_\theta^{*1}$ because $\mathcal{B}$ at that point only contains trajectories from random rollouts, which may contain uninformative and possibly detrimental behavior to regularize towards for certain exploration policy choices. In our implementation, the initial policies sample uniformly from actions in all states.

We describe in detail how the trajectory buffers $\mathcal{B}$ are built. Let $\mathcal{B}_i$ denote the trajectories gathered from training iteration $i$'s best response and empirical game updates. These are added to the trajectory buffer, $\mathcal{B} \leftarrow \mathcal{B} \cup \mathcal{B}_i$. Then, Ex$^2$PSRO filters $\mathcal{B}$ by taking the top $N$ trajectories that maximize criterion $C(\tau)$.

The Ex$^2$PSRO procedure is depicted schematically in Fig. 2. We define or train exploration policy $\pi_{ex}^i$. Next, we train our regularized best responses by optimizing (7) with SAC, augmenting its actor loss with the regularization term. Ex$^2$PSRO then updates the empirical game $\overline{\Gamma}$ and, subsequently, meta-strategies using the MSS $\mathcal{M}$. The regularization strength $\beta_i$ is initialized to $\beta_{init}$ and linearly annealed across $S_\beta$ PSRO iterations, where $\beta_{init}$ and $S_\beta$ are hyperparameters. This process repeats until the max number of Ex$^2$PSRO iterations is reached.

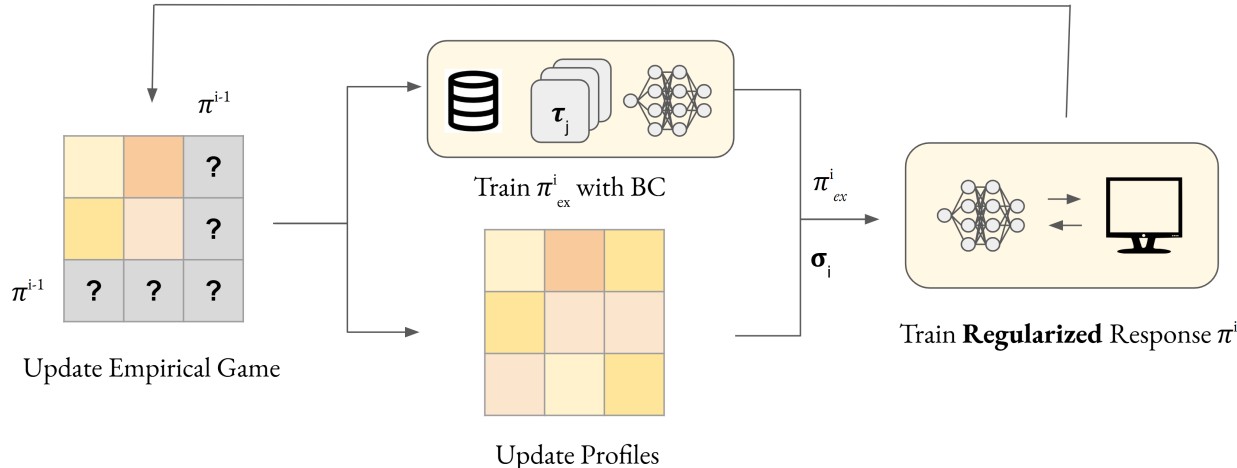

*Figure 2.* After the first iteration, Ex$^2$PSRO cycles through three steps: (1) updating the empirical game model $\overline{\Gamma}$, (2) deriving the response target $\mathcal{M}(\overline{\Gamma}) = \sigma^i$ using MSS $\mathcal{M}$ while training exploration policy $\pi_{ex}^i$ using trajectories $\{\tau_j\}_{j=1}^N$, and (3) training a new strategy $\pi_\theta^i$. Following steps (1) and (3), all observed simulation trajectories $\tau_j$ are added to buffer $\mathcal{B}$ and filtered using criterion $C(\tau_j)$.

## 5. Results

### 5.1. Environments

We first demonstrate how Ex$^2$PSRO operates in a hand-crafted dynamic environment (Fig. 3) exhibiting the coordination issue exemplified in Fig. 1. Players alternately choose to move left or right, and the episode ends after 10 total actions. There are two equilibria of interest. The first, which we refer to as Equilibrium 1, consists of players going between states $s_0$ and $s_1$, repeatedly gathering +1 reward. The second, analogously Equilibrium 2, has players reach and alternate between states $s_7$ and $s_8$, which collects lower rewards initially but dominates the first in overall return. Coordination is vital in achieving the latter equilibrium. The second equilibrium strategy is a best-response only to strategies similar to itself, meaning it cannot be found through strict best-response. Using a welfare-oriented objective does not help in this case.

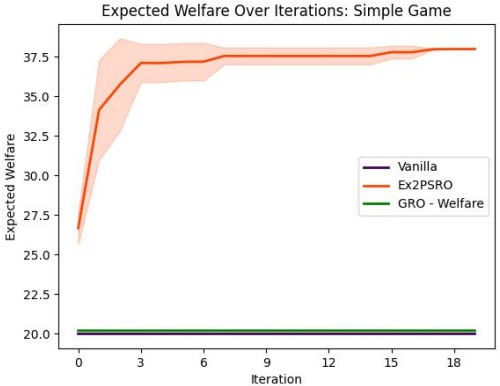

*Figure 4.* Welfare trends over PSRO iterations using Vanilla, Generalized Response Objectives with $\alpha = .5$ optimizing welfare, and Ex$^2$PSRO with $\beta_{init} = .1$. in the proof-of-concept MDP.

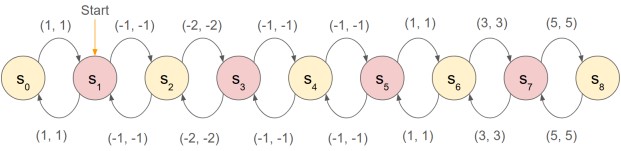

*Figure 3.* Hand-crafted Markov Decision Process

In this proof-of-concept MDP environment, Vanilla PSRO and GRO welfare do not add the strategy to move right towards $s_7$ and $s_8$ to the empirical game, therefore never discovering Equilibrium 2. They both achieve a welfare of 20.0. GRO Welfare, which optimizes welfare at each iteration, is inherently limited by its focus on myopic wel-

fare improvements. In contrast, Ex$^2$PSRO bookmarks high-welfare trajectories that travel to states $s_7$ and $s_8$, constructs strategies similar to such behavior, and eventually includes relevant strategies in $\overline{\Gamma}$ to discover Equilibrium 2. Ex$^2$PSRO achieves a significantly higher welfare of 38.0. This highlights the effectiveness of its regularization-driven approach in uncovering higher-welfare equilibria that are inaccessible through immediate or myopic optimization. Welfare plots for this experiment are shown in Fig. 4.

We more extensively test our approach in four game variants: two disparate versions of two benchmarks. *Harvest* (also known as *Gathering*) (Perolat et al., 2017) is a grid-world sequential social dilemma game where players collect apples that stochastically spawn, each of which gives +1

reward. After being collected, apples respawn at dedicated locations with probability proportional to the number of surrounding apples at each timestep. The players individually benefit by gathering as many apples as possible, but collectively benefit by leaving sufficient apples to enable respawning. Players may also shoot a laser, firing a beam several spaces ahead of them and removing any hit player from the game for a fixed number of time steps. We test our approach in two Harvest maps, referred to as Harvest-Dense and Harvest-Sparse, designed to be distinct enough to significantly change underlying game dynamics. We elaborate further in App. C.1.

We also test our approach on a 2-player, turn-based, imperfect information bargaining game, employing a version defined by Lewis et al. (2017). The game (which we refer to as *Bargaining*) features a pool of items that players must divide among themselves. Each player has a random, private value function that assigns a non-negative utility value to each item. Players alternate making offers of division until a player accepts or the game reaches a maximum number of iterations. If a player accepts, players receive utility according to the accepted asset allocations and their respective value functions, discounted by $\gamma^t$ where $0 < \gamma < 1$ is a hyperparameter and $t$ is the number of time steps so far. If no agreement is reached by the turn limit, both players receive zero utility. We test our approach in two Bargaining variants (detailed in App. C.2), setting $\gamma$ to .99 and .95. Discount factor differences *drastically* modify game dynamics, so experiments in both settings are essential for a thorough evaluation of Ex$^2$PSRO.

### 5.2. Implementation and Reproducibility

We build on an implementation of PSRO in an existing codebase, OpenSpiel, provided by Deepmind (Lanctot et al., 2019). Our experiments use an MSS called *regularized replicator dynamics* (RRD) (Wang & Wellman, 2023), chosen for its simplicity and strong empirical performance. RRD has a regret threshold parameter $\lambda$ that determines the regret at which RD terminates, yielding a $\lambda$-Nash equilibrium. $\lambda$ is linearly annealed across all PSRO iterations. When $\lambda = 0$, RRD reduces to RD.

We compare Ex$^2$PSRO to vanilla PSRO, a SOTA game-solving framework. For Ex$^2$PSRO, we performed a grid search over $\lambda$ and $\beta_{init}$, keeping $S_\beta = 25$ constant, where each parameter setting's welfare is averaged over 5 trials. Then, we select the parameter setting ($\lambda$, $\beta_{init}$) with the highest welfare and provide it an additional 5 trials, totaling 10 trials for the final result. Vanilla PSRO is tuned over $\lambda$, where we immediately run 10 trials for each $\lambda$ candidate and report data from the highest-performing parameter setting. Due to computational limitations (App. G), Ex$^2$PSRO regret values are averaged over 5 trials, while all other settings are

averaged over 10. Details on hyperparameters, tuning, and implementation are provided in Apps. A.1, B, C.1, and C.2.

### 5.3. Varying Exploration Policy

We also conduct experiments varying Ex$^2$PSRO's exploration policy to analyze its dependence on the target behavior it regularizes towards. Table 4 in App. A.1 summarizes the four tested exploration policy alternatives. Uniform sets $\pi_{ex}^i$ to a policy that selects actions uniformly for all states. MinWelfare and MaxWelfare use BC to train $\pi_{ex}^i$ on datasets generated by trajectory criteria prioritizing low and high welfare, respectively, and disregarding return distribution among players. PrevBR sets $\pi_{ex}^i$ to the most recently added strategy. From this point forward, we refer to this suite of ablations over $\pi_{ex}^i$ as "Ablation-$\pi_{ex}$" experiments.

All Ablation-$\pi_{ex}$ experiments use the same regret threshold $\lambda$ and initial regularization $\beta_{init}$ as tuned Ex$^2$PSRO, similarly fixing $S_\beta = 25$. This experiment isolates and measures to what extent Ex$^2$PSRO's performance depends on regularization choice versus regularization strength. Ablations over $\beta_{init}$ and $\lambda$ are provided in App. H.2.

### 5.4. Equilibrium Welfare

Fig. 5 shows welfare of equilibria returned from tuned versions of vanilla PSRO, Ex$^2$PSRO, and Ablation-$\pi_{ex}$.

Welfare differences should be measured relative to the game's distribution of available equilibria, though computing this distribution becomes computationally intractable for large games. The bar graph in Fig. 5 uses normalized values to approximate this relative measurement, providing a meaningful representation of the improvements achieved.

We also provide significance tests for our welfare results, comparing Ex$^2$PSRO to vanilla PSRO and Ablation-$\pi_{ex}$. We use Welch's t-test to calculate P-values as informed by Colas et al. (2019) for its robustness to false positives compared to non-parametric tests. P-values are not adjusted for multiple hypotheses.

Performance in these four benchmarks indicates that Ex$^2$PSRO tends to increase welfare through strategy exploration, yielding welfare improvements over vanilla PSRO with low P-values on all benchmarks tested. Ex$^2$PSRO also tends to outperform approaches in Ablation-$\pi_{ex}$ with low P-value, noting a few exceptions in the next paragraph.

While Ex$^2$PSRO and Uniform showed comparable performance in Harvest benchmarks, Ex$^2$PSRO demonstrated distinct advantages in Bargaining scenarios. Similarly, P-values indicate Ex$^2$PSRO and MaxWelfare performed similarly in Harvest-Sparse. Performance across environments reveals key relationships between game dynamics and regularization. In Ablation-$\pi_{ex}$, Harvest's welfare improve-

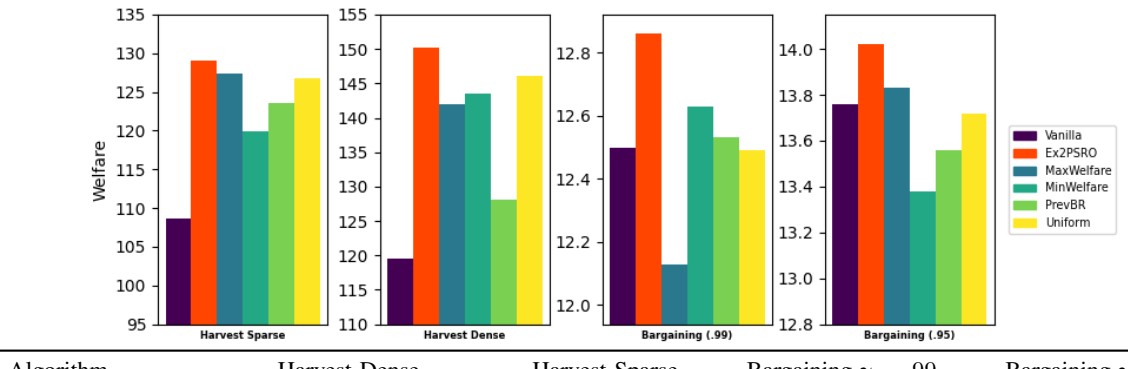

| Algorithm | Harvest-Dense | Harvest-Sparse | Bargaining $\gamma = .99$ | Bargaining $\gamma = .95$ |
|---|---|---|---|---|
| Ex$^2$PSRO (Ours) | **150.2 $\pm$ 7.9** \| N/A | **129.0 $\pm$ 7.5** \| N/A | **12.86 $\pm$ 0.26** \| N/A | **14.02 $\pm$ 0.22** \| N/A |
| Vanilla | 119.6 $\pm$ 3.4 \| $p < .01$ | 108.6 $\pm$ 8.9 \| $p < .01$ | 12.50 $\pm$ 0.55 \| $p = .04$ | 13.76 $\pm$ 0.34 \| $p = .03$ |
| Uniform | 146.0 $\pm$ 8.3 \| $p = .13$ | 126.8 $\pm$ 6.3 \| $p = .24$ | 12.49 $\pm$ 0.33 \| $p = .01$ | 13.72 $\pm$ 0.34 \| $p = .02$ |
| MinWelfare | 143.5 $\pm$ 9.5 \| $p = .05$ | 119.9 $\pm$ 10.1 \| $p = .02$ | 12.63 $\pm$ 0.31 \| $p = .05$ | 13.38 $\pm$ 0.73 \| $p = .01$ |
| MaxWelfare | 142.0 $\pm$ 6.1 \| $p = .01$ | 127.4 $\pm$ 7.6 \| $p = .32$ | 12.13 $\pm$ 0.37 \| $p < .01$ | 13.83 $\pm$ 0.48 \| $p = .08$ |
| PrevBR | 128.2 $\pm$ 4.3 \| $p < .01$ | 123.4 $\pm$ 7.3 \| $p = .05$ | 12.53 $\pm$ 0.33 \| $p = .01$ | 13.56 $\pm$ 0.39 \| $p < .01$ |

*Figure 5.* Welfare of equilibria found by vanilla PSRO, Ex$^2$PSRO, and Ablation-$\boldsymbol{\pi_{ex}}$ with significance test results. These tests *alternatively hypothesize whether Ex$^2$PSRO yields higher welfare equilibria than vanilla PSRO and Ablation-$\boldsymbol{\pi_{ex}}$*. Bolded values in the table indicate each environment's highest welfare-yielding approach. Welfare is averaged over 10 trials with tuned hyperparameters ($\boldsymbol{\lambda}$, $\boldsymbol{\beta_{init}}$). Bar graph axis bounds are scaled to display the middle 68% of welfare found across all trials (185 per game), discussed in App. D. This is intended to provide a normalized view of observed differences, further discussed in §5.4 and App. D. Error bars are omitted for clarity.

*Table 1.* Combining Ex$^2$PSRO with GRO. P-Values of Ex$^2$PSRO and GRO are measured with respect to Vanilla. P-Values of GRO+Ex$^2$PSRO are measured with respect to GRO.

| Algorithm | Harvest-Dense | Harvest-Sparse | Bargaining $\gamma = .99$ | Bargaining $\gamma = .95$ |
|---|---|---|---|---|
| Vanilla | 119.6 $\pm$ 3.4 | 108.6 $\pm$ 8.9 | 12.50 $\pm$ .55 | 13.76 $\pm$ .34 |
| GRO | 134.6 $\pm$ 9.1 | 115.2 $\pm$ 11.0 | 12.99 $\pm$ .33 | 15.07 $\pm$ .10 |
| Ex$^2$PSRO (Ours) | 150.2 $\pm$ 7.9 | **129.0 $\pm$ 7.5** | 12.86 $\pm$ .26 | 14.02 $\pm$ .22 |
| GRO + Ex$^2$PSRO (Ours) | **151.0 $\pm$ 6.2** \| $\boldsymbol{p < .01}$ | 122.3 $\pm$ 9.5 \| $p = .07$ | **13.49 $\pm$ .39** \| $\boldsymbol{p < .01}$ | **15.24 $\pm$ .08** \| $\boldsymbol{p < .01}$ |

ments naturally emerge from any regularization that reduces immediate apple collection, allowing more apples to spawn. Shown with detailed significance tests in App. D, all Ablation-$\pi_{ex}$ variants achieved improvements over vanilla PSRO in Harvest with low P-values. Ex$^2$PSRO particularly shines in Bargaining, where achieving high-welfare behaviors requires navigating more nuanced and complex interactions.

Fig. 6 demonstrates Ex$^2$PSRO's ability to discover high-welfare solutions more rapidly than other tested approaches. While Uniform exploration achieves quick initial gains in Harvest, its progress plateaus early. MinWelfare shows swift improvements in Bargaining $\gamma = .99$ but gradual in Harvest. MaxWelfare welfare trends vary widely while PrevBR's welfare increases slowest within Ablation-$\pi_{ex}$. These results underscore a key strength of Ex$^2$PSRO: its strong performance (in its consistency, speed, and magnitude of welfare increases) can be attributed to both the presence and deliberate choice of regularization.

Additional analysis and data for hyperparameter searches over $\lambda$ and $\beta_{init}$ are provided in Apps. H, H.2, and Table 8, while ablation experiments varying $\beta_{init}$ or $\lambda$ are detailed in App. H.2. Ex$^2$PSRO's tuning was conservative, potentially missing stronger parameter settings, whereas vanilla PSRO underwent exhaustive tuning, and results reflect its highest-welfare configuration.

### 5.5. Combining with Generalized Response Objectives

In principle, any PSRO variant can incorporate Ex$^2$PSRO by including the regularization term (Eq. 5) in its response objective. We demonstrate Ex$^2$PSRO's composability by combining it with another approach addressing high-welfare equilibrium selection: Generalized Response Objectives (GRO) (Wang & Wellman, 2024). We highlight that GRO is not considered a competing approach but one whose performance can be bolstered and amplified by Ex$^2$PSRO.

Experiments augmenting vanilla PSRO and GRO with Ex$^2$PSRO are shown in Table 1. With low P-values, the

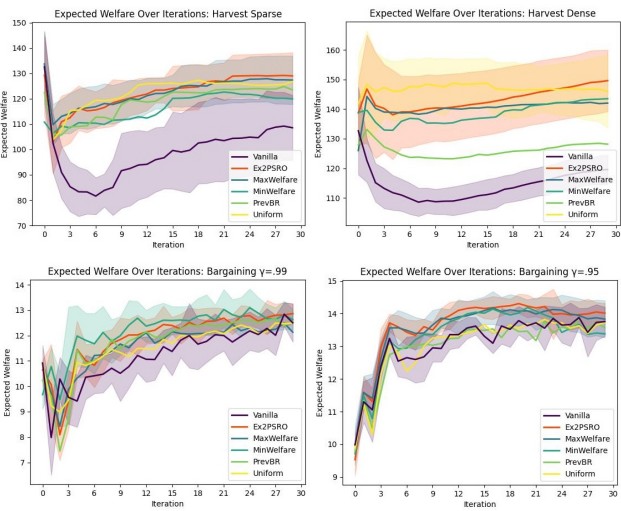

*Figure 6.* Welfare plots for vanilla PSRO, Ex$^2$PSRO, and Ablation-$\pi_{ex}$. We include error bars for only vanilla PSRO, Ex$^2$PSRO, and the top performer in Ablation-$\pi_{ex}$ for each game for clarity. App. H has additional plots.

data supports that combining Ex$^2$PSRO with GRO tends to find higher welfare equilibria than GRO alone. Compared to tested approaches, GRO+Ex$^2$PSRO tends to find the highest welfare equilibria, suggesting that the combination amplifies welfare improvements. An exception is Harvest-Sparse, where Ex$^2$PSRO achieved the highest welfare. Note that we do not compare GRO and Ex$^2$PSRO directly nor tune GRO-specific parameters. Instead, these experiments support Ex$^2$PSRO's flexibility and composability with different PSRO variants to improve welfare potential. App. A.1 provides additional details on hyperparameters and procedures.

**5.6. Regret $\times$ Welfare Analysis**

The regret and welfare of final profiles discovered are shown in Fig. 7, where square markers indicate the averaged performance of a particular algorithm. In Bargaining $\gamma = .99$ and Harvest-Sparse, Ex$^2$PSRO dominates all tested approaches. While Ex$^2$PSRO generally tends to find high-welfare solutions with lower regret, Uniform and PrevBR tend to find slightly lower regret profiles than Ex$^2$PSRO in Bargaining $\gamma = .95$, In Harvest-Dense, MaxWelfare finds lower regret profiles than Ex$^2$PSRO.

Small positive correlations between regret and welfare in Harvest could reflect the game structure of social dilemmas, where high-welfare profiles tend to have high-value strategic deviations. In Bargaining, there is a negative correlation between the two metrics, suggesting relative stability of high-welfare solutions found.

Next, we compare and analyze regret trends of intermediate profiles of all approaches. Details on true game regret

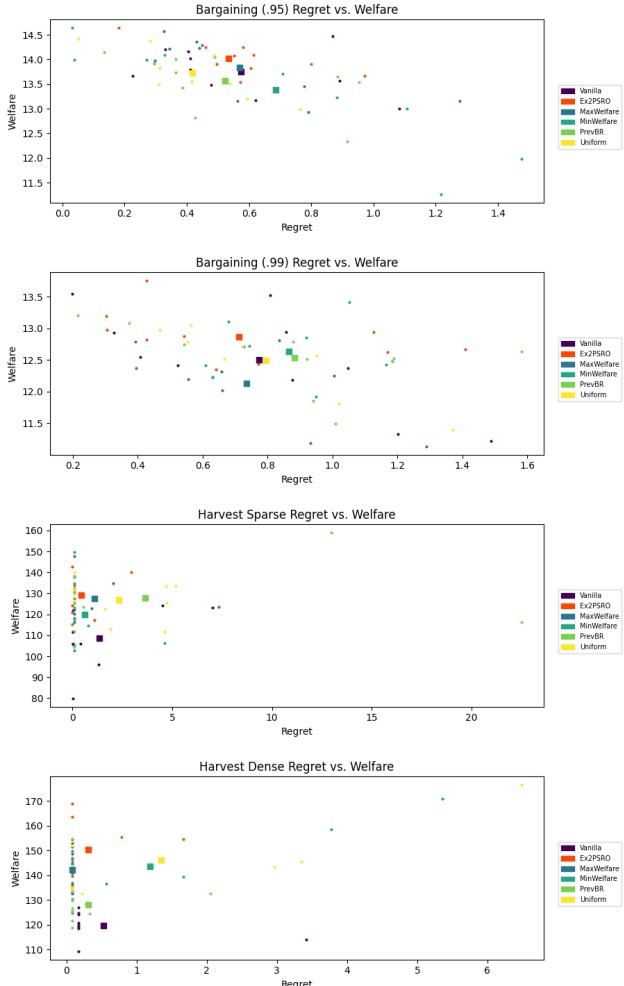

*Figure 7.* Final regret and welfare of output solutions across approaches. Circles and squares represent individual trials and averaged performance within a specified approach, respectively.

approximations are provided in App. G. Trends in Fig. 8 and App. H indicate that Ex$^2$PSRO appropriately generates approximate equilibria in all benchmarks. We note that all regret trends approach equilibria as shown by the untruncated plots (App. H) but Fig. 8 enlarges some of the last few iterations' datapoints for clarity.

Ex$^2$PSRO tends to find lower regret solutions faster than vanilla PSRO, indicating a link between convergence and regularization. Relative to Ablation-$\pi_{ex}$ experiments, Ex$^2$PSRO is consistently among the top performers in convergence speed and final solution regret. These results suggest Ex$^2$PSRO's presence and choice of regularization together provide a mechanism to improve convergence while encouraging the discovery of solutions with higher welfare and, possibly, lower regret. We highlight that Ex$^2$PSRO's true regret estimation was more thorough than other ap-

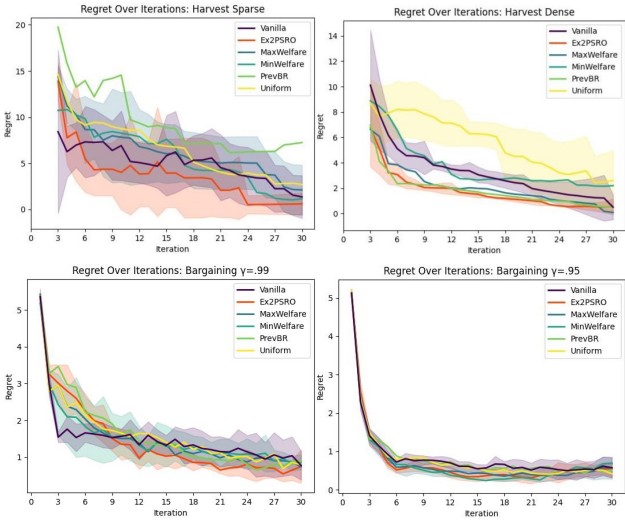

*Figure 8.* Regret plots for vanilla PSRO, Ex²PSRO, and Ablation-$\pi_{ex}$. We include error bars for only vanilla PSRO, Ex²PSRO, and the top performer in Ablation-$\pi_{ex}$ for clarity. Harvest plots observe trends in later iterations. App. H has additional plots.

proaches (App. G), meaning vanilla and Ablation-$\pi_{ex}$ solutions are more likely to have even higher true regret than reported. In summary, regret trends favor Ex²PSRO despite being given a disadvantage in terms of its regret evaluation. Additional regret trends and ablations are provided in Apps. H and H.2.

## 6. Conclusions

We investigated the possibility of steering MARL game-solving toward preferred equilibria, particularly those with higher welfare. Toward this end, we introduce the concept of *exploration policies*. To construct these exploration policies, Ex²PSRO filters already-present simulation traces for high-welfare yielding behavior and applies behavior cloning to the filtered data. Then, Ex²PSRO uses these policies to regularize best-response computations towards promising behavior within the PSRO framework. Notably, since Ex²PSRO does not require any additional simulations relative to PSRO, it has shown to be a simple yet effective extension to the current SOTA. We tested our approach in two benchmark environments: a sequential social dilemma game and a game of bargaining over resource division.

Our experiments show that Ex²PSRO tends to improve equilibrium welfare across all benchmarks. By varying the exploration policy, we show that Ex²PSRO's performance hinges not only on its added regularization but also its choice of regularizing behavior. In our experiments, Ex²PSRO most consistently discovers lower-regret, higher-welfare solutions quicker than its vanilla counterpart and other exploration policy alternatives. These findings suggest that Ex²PSRO

provides a simple and effective mechanism to guide strategy exploration in service of high-welfare equilibrium selection. Experiments combining Ex²PSRO and Generalized Response Objectives support its composability with other PSRO-adjacent approaches to improve welfare. Lastly, experiments illuminate how a game's inherent structure may influence which regularization choices are beneficial or detrimental to performance.

## Impact Statement

This paper presents work whose goal is to advance the field of Machine Learning. There are many potential societal consequences of our work, none which we feel must be specifically highlighted here.

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

## A. Hyperparameters

All Soft Actor-Critic (SAC) parameters considered and chosen for each environment are shown in Table 2. The top section lists parameters held fixed while the bottom lists parameters included in the parameter search. These chosen hyperparameters were common across vanilla PSRO, Ex$^2$PSRO, and Ablation-$\pi_{ex}$.

*Table 2.* Soft-Actor Critic Parameter Search

| Parameter | Candidates | Harvest | Bargaining |
|---|---|---|---|
| Network Depth | [2] | 2 | 2 |
| Activation | [ReLU] | ReLU | ReLU |
| Discount Factor | [.99] | .99 | .99 |
| Gradient Steps per Update | [1] | 1 | 1 |
| BC Epochs | [50] | 50 | 50 |
| BC Learning Rate | [3e-4] | 3e-4 | 3e-4 |
| BC Batch Size | [256] | 256 | 256 |
| BC Network Depth | [2] | 2 | 2 |
| BC Network Width | [Match Network Width] | 50 | 100 |
| BC Entropy Threshold | None | .2 | .8 |
| Max Trajectory Buffer Size | None | 1000 | 100,000 |
| Entropy Temperature ($\alpha$) | [0, 1e-3, 3e-3, 1e-2, 3e-2, 1e-1] | 3e-2 | 1e-1 |
| Value Epsilon Clip ($\epsilon_v$) | [.2 .5 1.0 2.0] | .2 | .2 |
| Update Every ($P$) | [1, 2, 5, 10] | 2 | 1 |
| Number of Training Steps ($S$) | [100K, 500K, 1M 2M 3M 4M] | 2M | 100K |
| Policy Learning Rate ($\lambda_p$) | [3e-5, 1e-4, 3e-4] | 3e-5 | 3e-5 |
| Value Learning Rate ($\lambda_v$) | [3e-5, 1e-4, 3e-4] | 1e-4 | 3e-4 |
| Max Buffer Size ($B$) | [50K 100K 200K 500K] | 100K | 50K |
| Network Width ($W$) | [50 100 250] | 50 | 100 |
| Batch Size ($b$) | [32, 64] | 64 | 64 |

We divided the parameter search for SAC hyperparameters into three phases, each of which advanced a subset of parameters to the next. The first parameter search was over high-priority parameters that we believed would highly influence performance. The second search was over lower-priority parameters that we believed were important when fine-tuning policies. The evaluation of a policy in rounds 1 and 2 was dictated by the average final policy returns across 3 trials when responding to a uniform action policy. In round 3, we compared performance between the strongest configurations determined from the previous two rounds, averaging across 5 trials responding to a uniform mixture of policies. The policy mixture was generated by five iterations of PSRO using SAC parameters from one of the parameter candidates in round 3. If parameter configurations resulted in similar performance, configurations with less computational cost were favored. These details are summarized in Table 3.

*Table 3.* Hyperparameter Tuning Procedure Summary

| Round | Parameters | Trials | Response Target |
|---|---|---|---|
| 1 | $\alpha, \epsilon_v, P, S$ | 3 | $\pi_\theta = U(\mathcal{A})$ |
| 2 | $\lambda_p, \lambda_v, B, W, b$ | 3 | $\pi_\theta = U(\mathcal{A})$ |
| 3 | All finalists | 5 | $\sigma = U(\pi_\theta^1, \ldots, \pi_\theta^5)$ |

Table 2 also includes some parameters for Ex$^2$PSRO, specifically behavior cloning parameters and trajectory buffer size. These parameters were not included in the search, the latter of which was chosen based on initial empirical results. Future work could be dedicated to determining how trajectory buffer size affects Ex$^2$PSRO's performance.

In Table 5, we show the PSRO and Ex$^2$PSRO regularization-related hyperparameters used to generate the welfare values in Fig. 5, where the candidate hyperparameters search over are shown in Tables 8. The number of steps to anneal exploration regularization $S_\beta$ was held fixed at 25. The Ex$^2$PSRO parameter search was conducted over two parameters: RRD

regularization $\lambda$ and initial exploration regularization $\beta_{init}$. Ex$^2$PSRO parameters were selected by conducting 5 trials for each parameter configuration. For each environment, the setting yielding equilibria with the highest welfare was chosen and given an additional 5 trials to produce the results in Fig. 5. Vanilla PSRO was tuned across only $\lambda$ as it does not include exploration regularization. Each $\lambda$ was given 10 trials and we immediately report data from the best-performing parameter setting in Fig. 5.

### A.1. Generalized Response Objectives

GRO optimizes a modified response objective at each iteration of PSRO. Our experiments optimized social welfare, where the resultant policy optimizes the following, with slight notation changes:

$$\pi_\theta^i \approx \arg\max_{\pi_\theta} \mathbb{E}_{\pi_\theta|i} \sum_{t=1}^{T} (\alpha r_t^p + (1-\alpha)r_t^{-p}) = \arg\max_{\pi_\theta} J_{GRO}(\theta, \alpha),$$

where superscript $p$ and $-p$ indicate rewards assigned to player $p$ and all others except $p$, respectively, and $\alpha$ is a hyper-parameter. We selected $\alpha = .8$ as justified by the original work (Wang & Wellman, 2024). When combining GRO and Ex$^2$PSRO, behavior regularization was also applied, meaning the aggregate optimization is the following:

$$\pi_\theta^i \approx \arg\max_{\pi_\theta}(J_{GRO}(\theta, \alpha) - \beta_i R^i(\theta))$$

Experiments combining GRO and Ex$^2$PSRO were not tuned across hyperparameters. Ex$^2$PSRO's $\beta_{init}$ was fixed to .01 and $\lambda$ was fixed to .5 and 5 in both Bargaining and Harvest variants, respectively.

_Table 4._ Ex$^2$PSRO vs. Ablation-$\pi_{ex}$

| Name | Criterion | Exploration Policy |
|------|-----------|--------------------|
| Ex$^2$PSRO | Eq. 2 | Eq. 3 |
| Ablation-$\pi_{ex}$ | | |
| Uniform | N/A | U($\mathcal{A}$) |
| MinWelfare | $-\sum_p V_j^p$ | Eq. 3 |
| MaxWelfare | $\sum_p V_j^p$ | Eq. 3 |
| PrevBR | N/A | $\pi_\theta^{i-1}$ |

_Table 5._ Vanilla PSRO and Ex$^2$PSRO Parameter Summary

| Game | Parameter | PSRO | Ex$^2$PSRO |
|------|-----------|------|------------|
| Harvest-Dense | RRD ($\lambda$) | 10 | 5 |
| | $\beta_{init}$ | - | 1e-2 |
| Harvest-Sparse | RRD ($\lambda$) | 2 | 5 |
| | $\beta_{init}$ | - | 1e-2 |
| Bargaining $\gamma = .99$ | RRD ($\lambda$) | .1 | .2 |
| | $\beta_{init}$ | - | 3e-4 |
| Bargaining $\gamma = .95$ | RRD ($\lambda$) | .5 | .5 |
| | $\beta_{init}$ | - | 1e-2 |

## B. Behavior Cloning

Exploration policies $\pi_{ex}^i$ at each iteration are trained using behavior cloning (BC) on a trajectory buffer $\mathcal{B}$ that consists of trajectories maximizing a criterion $C(\tau_j)$ as noted in Eq. 2. The training of exploration policy $\pi_{ex}^i$ is be executed

immediately before training responses $\pi_\theta^i$ at each iteration $i$.

The full behavior cloning algorithm is shown in Algorithm 2, where we train a multilayer perceptron (ReLU activations) with a softmax output layer using cross-entropy loss. Batch sizes, number of epochs, size of trajectory buffer $\mathcal{B}$, and learning rates are enumerated in Table 2. We insert an early stopping criterion (called "BC Entropy Threshold" in Table 2) to avoid divide-by-near-zero errors when calculating the regularization term in Equation 5.

---

**Algorithm 2** Create Exploration Policy with Behavior Cloning

---

**Inputs**: Trajectory data $\mathcal{B}$, Epochs $E$, Minibatch size $M$, learning rate $\alpha$, model shape $S$
**Outputs**: Exploration policy $\pi_{ex}^i$ parameterized by $\theta_{ex}^i$

  1: Initialize deep model $\pi_{ex}^i$ of shape $S$ and parameters $\theta_{ex}^i$
  2: Create dataloader $\mathcal{D}$ using data $\mathcal{B}$ of minibatch size $M$
  3: **for** $E$ epochs **do**
  4:     **for** each minibatch $(o_k, a_k)$ in $\mathcal{D}$ **do**
  5:         $J(\theta_{ex}^i) = CE(\pi_{ex}^i(o_k), a_k)$
  6:         $\theta_{ex}^i = \theta_{ex}^i - \alpha \nabla J(\theta_{ex}^i)$
  7:     **end for**
  8:     Shuffle and rebatch dataloader $\mathcal{D}$
  9: **end for**
10: **return** $\pi_{ex}^i$

---

## C. Environment Details

### C.1. Harvest Details

Each space in the *Harvest* (Perolat et al., 2017) grid-world is characterized by one of four states: empty, wall, apple, or a player. Apples are only able to reside in dedicated locations. Players initially spawn randomly in one of the four corners. The grid worlds are shown in Fig. 9.

A player's observation is defined by their position and orientation. In this implementation, agents observe a rectangular viewing box of 7 spaces forward and 3 to the left and right, both of which include their position. The vectorized state input to a policy neural network $\pi_\theta$ is created by unraveling the viewing box into a length 21 tensor, creating 4 indicator tensors for each of the possible space states (empty, wall, apple, player). These tensors are then concatenated to create an aggregate length 84 tensor. The player tensor has a value of -1 to refer to itself, 1 for other players, and 0 for spaces without other players.

At each time step, players take actions simultaneously, where each can take one of eight actions: {left, right, forward, backward, turn clockwise, turn counter-clockwise, laser, no operation}. The first four change the player's location by moving relative to their orientation. The next two modify their orientation. The "laser" action fires a beam that spans the player's viewing box (3x7) and removes hit players from the game for 25 time steps.

If an apple spawn location is empty (no player and no apple), an apple respawns with probability proportional to the number of apples within 2 units of Euclidean distance. Table 6 summarizes these respawn probabilities. An episode terminates when either there are no apples on the map or 1000 time steps are reached.

We construct two variants of the Harvest game, which we term **Harvest-Dense** and **Harvest-Sparse**. The former's map is densely populated with apples and has initial spawn points that are close to the players' spawn points. The latter's map consists of a central trove of apples, forcing players to interact with each other in the middle. Harvest-Sparse can be considered a more challenging variant than Harvest-Dense, as the central allocation and decreased presence of apples make coordination more important for high welfare.

*Table 6.* Harvest Respawn Probabilities

| Apples in Vicinity (2 units) | 0 | 1 | 2 | $\geq 3$ |
|---|---|---|---|---|
| Respawn Probability | 0.000 | .005 | .020 | .050 |

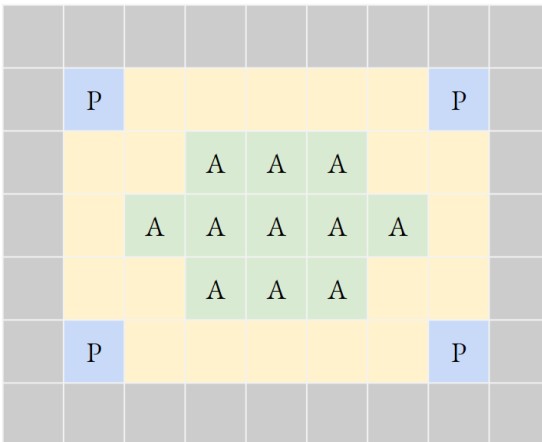 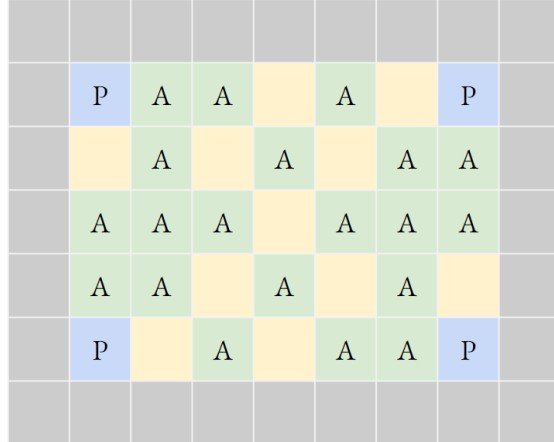

*Figure 9.* Diagrams of the Harvest Sparse and Harvest Dense environments. Blue and green tiles indicate where players and apples can spawn, respectively. Gray tiles are walls and yellow tiles are empty.

### C.2. Bargaining Details

The objective in Bargaining (Lewis et al., 2017) is for players to divide up the items so that all items are owned by one of the 2 players. Each player's random, private value function (assigns a non-negative utility value to each of the items) is constrained such that: (1) the sum of the utility values of all items is the same for both players (i.e., the individual values per item may be different between users but the sum of the items' utilities is the same for both players); (2) each item has a non-zero value for at least one user; and (3) some items have a non-zero value for both users. The game starts by having one player offer a certain division of assets.[4] The other player can either accept the offer or decline and present a counteroffer. This process continues until all assets have been divided or a turn limit has been reached. Both players receive zero utility if no agreement is reached by the turn limit. In our implementation, the total turn limit is 10, which includes moves by both players. Received utilities are discounted after each iteration by $\gamma$. Smaller $\gamma$ leads players to strategize more myopically, focusing on shorter-term gains rather than long-term. This changes game dynamics dramatically as players may prioritize making deals in less time-steps at the cost of a potentially higher-welfare yielding division of assets. Our implementation sets $\gamma = .99$ and $\gamma = .95$. This prevents players from forcing ultimatums at the last iteration, a relatively degenerate version of the game. We adapt an implementation provided by OpenSpiel to establish symmetry between players by having players sample from the same pool of valuations and determining the starting player by a coin flip. We defer further implementation details to Deepmind's OpenSpiel repository (Lanctot et al., 2019).

## D. Welfare Distributions and Significance Tests

We underscore that welfare improvement should be measured relative to the distribution of welfare values of available equilibria in each game. To that end, raw welfare values are not as valuable to examine. For example, suppose a game contains equilibria whose welfare values all lie within [1000, 1005]. While an improvement from 1000 to 1004 may not seem large relative to the raw welfare scores, it is large relative to the distribution of equilibria welfare.

Calculation of this actual distribution is infeasible, as it depends on finding all equilibria. To capture the proportions relative to what we can compute, for Fig. 5 we scale the y-axis to cover the inner 68% of points we found across all runs of each game. This is done to encourage analysis with respect to the welfare distribution in the game, but we do not claim that our data samples accurately reflect the distribution of available equilibria. Fig. 7 provides significance tests comparing all pairs of tested approaches listed in Fig. 5.

---

[4]The starting player is determined by a coin flip, establishing symmetry in the game.

*Table 7.* All P-values use Welch's t-test, verifying whether the approaches listed in the row outperform those in the column in the listed game benchmark.

| | Harvest-Dense | | | | | |
| --- | --- | --- | --- | --- | --- | --- |
| | Vanilla | Ex$^2$PSRO | Uniform | MinWelfare | MaxWelfare | PrevBR |
| Vanilla | N/A | 1.00 | 1.00 | 1.00 | 1.00 | 1.00 |
| Ex$^2$PSRO | 4e-8 | N/A | .131 | .052 | .009 | 1e-6 |
| Uniform | 4e-7 | .869 | N/A | .269 | .118 | 1e-5 |
| MinWelfare | 5e-6 | .948 | .731 | N/A | .340 | 2e-4 |
| MaxWelfare | 4e-8 | .991 | .882 | .660 | N/A | 1e-5 |
| PrevBR | 6e-5 | 1.00 | 1.00 | 1.00 | 1.00 | N/A |
| | Harvest-Sparse | | | | | |
| | Vanilla | Ex$^2$PSRO | Uniform | MinWelfare | MaxWelfare | PrevBR |
| Vanilla | N/A | 1.00 | 1.00 | 1.00 | 1.00 | 1.00 |
| Ex$^2$PSRO | 1e-5 | N/A | .243 | .018 | .321 | .054 |
| Uniform | 3e-5 | .757 | N/A | .043 | .575 | .139 |
| MinWelfare | 8e-3 | .982 | .957 | N/A | .961 | .806 |
| MaxWelfare | 4e-5 | .679 | .425 | .039 | N/A | .123 |
| PrevBR | 3e-4 | .946 | .860 | .194 | .877 | N/A |
| | Bargaining $\gamma = .99$ | | | | | |
| | Vanilla | Ex$^2$PSRO | Uniform | MinWelfare | MaxWelfare | PrevBR |
| Vanilla | N/A | .958 | .481 | .737 | .048 | .558 |
| Ex$^2$PSRO | .042 | N/A | .006 | .045 | 1e-4 | .012 |
| Uniform | .519 | .994 | N/A | .829 | .017 | .605 |
| MinWelfare | .263 | .955 | .171 | N/A | .002 | .247 |
| MaxWelfare | .952 | 1.00 | .983 | .998 | N/A | .990 |
| PrevBR | .442 | .988 | .395 | .753 | .010 | N/A |
| | Bargaining $\gamma = .95$ | | | | | |
| | Vanilla | Ex$^2$PSRO | Uniform | MinWelfare | MaxWelfare | PrevBR |
| Vanilla | N/A | .970 | .398 | .080 | .644 | .119 |
| Ex$^2$PSRO | .030 | N/A | .016 | .011 | .080 | .003 |
| Uniform | .602 | .984 | N/A | .103 | .719 | .171 |
| MinWelfare | .920 | .989 | .897 | N/A | .938 | .748 |
| MaxWelfare | .356 | .862 | .281 | .062 | N/A | .092 |
| PrevBR | .881 | .997 | .829 | .252 | .908 | N/A |

# E. Compute Resources

Experiments were run on the University of Michigan's research computing cluster. We summarize computational resources allocated for all Harvest and Bargaining variants. Using a CPU, each PSRO, Ex$^2$PSRO, and Ablation-$\pi_{ex}$ run took anywhere between 2 and 4 days to finish as there were a total of 30 RL policies being trained in sequence. Ex$^2$PSRO variants required roughly 8–12 GB of RAM to support trajectory buffer storage, which could be reduced through code changes allowing disk storage. Vanilla PSRO was run with 5 GB of RAM while Ex$^2$PSRO and Ablation-$\pi_{ex}$ runs were run with 12–15 GB of RAM. Additional computation was allocated for regret calculations, elaborated in App. G, which took an additional 3–4 days and 5 GB of RAM. For each game variant, experiments included 40 vanilla PSRO runs (tuning across $\lambda$), 105 Ex$^2$PSRO runs (100 for tuning across ($\lambda$, $\beta_{init}$), 5 for the finalist) and 40 Ablation-$\pi_{ex}$ runs (10 for each variant), totaling 185 runs per game. With 4 benchmarks, a total of 740 trials were conducted throughout, each requiring the aforementioned compute.

## F. Trajectory Buffer Sharing

A key part of the Ex$^2$PSRO algorithm is the trajectory buffer, which leverages simulation data generated during the PSRO process to guide the algorithm towards discovering higher-welfare equilibrium. This approach attributes additional value to trajectory data that would otherwise be discarded in vanilla PSRO after response training. Here, we discuss the assumption that trajectories are shared across players.

We emphasize that a shared trajectory buffer does not change the underlying nature of the game. Crucially, Ex$^2$PSRO still enforces that players only have access to their own information when making decisions. Ex$^2$PSRO's design is to be interpreted from the perspective of a game analyst: Ex$^2$PSRO is used to discover equilibria, not reflect how players would develop or change their strategies over time to arrive at an equilibrium. More concretely, access to a shared trajectory buffer is used to aid algorithmic discovery of higher-welfare equilibria, not advocating that players in the real world would have access to such a buffer for strategy generation.

## G. Estimating True Regret

Let $\Pi_T$ and $\Pi'_T$ be the un-regularized and regularized sets of strategies, respectively, at the end of a PSRO trial where $T$ is the total number of PSRO iterations. Note that for vanilla PSRO, $\Pi'_T = \emptyset$ and for Ex$^2$PSRO, $|\Pi_T| = T - S_\beta$ and $|\Pi'_T| = S_\beta$. To ensure Ex$^2$PSRO was rigorously evaluated, we augment Ex$^2$PSRO's $\Pi_T$ after each full run by training un-regularized best responses $\pi_\theta^{*i}$ for each profile $\sigma^i$ for which a regularized response $\pi_\theta^i \in \Pi'_T$ was created. We estimate the true game regret as the max deviation gain from the iteration's profile where we consider strategies in $\Pi_T \cup \Pi'_T$ and lower-bound the estimate by 0:

$$Regret(\sigma^i) = \max_{\pi \in \Pi_T \cup \Pi'_T} \max(u(\pi, \sigma^i_{-1}) - u(\sigma^i), 0). \tag{8}$$

All utilities are estimated empirically using a fixed number of simulations. We note that for Ex$^2$PSRO, $|\Pi_T \cup \Pi'_T| = T + S_\beta$ while for vanilla PSRO, $|\Pi_T \cup \Pi'_T| = T$. Ex$^2$PSRO's regret evaluation set includes $T$ non-regularized responses (just like vanilla PSRO) but contains an additional $S_\beta$ regularized responses. This discrepancy makes Ex$^2$PSRO's regret calculations more thorough than that of vanilla PSRO, meaning vanilla PSRO's regret has a higher chance of being underestimated than Ex$^2$PSRO. In other words, vanilla PSRO is given an "advantage" in terms of regret calculations. Despite this, regret values for Ex$^2$PSRO are equal to, arguably lower than, vanilla PSRO's in all benchmarks shown in §5.6. Trials in Ablation-$\pi_{ex}$ were not given non-regularized responses, meaning $\Pi_T = \emptyset$ and reported regret for Ablation-$\pi_{ex}$ trials similarly have higher chances of being underestimated.

## H. Additional Results

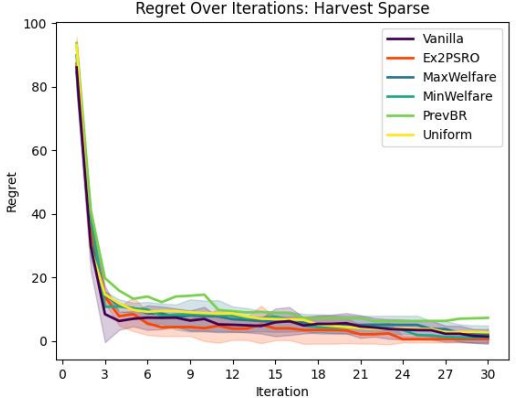 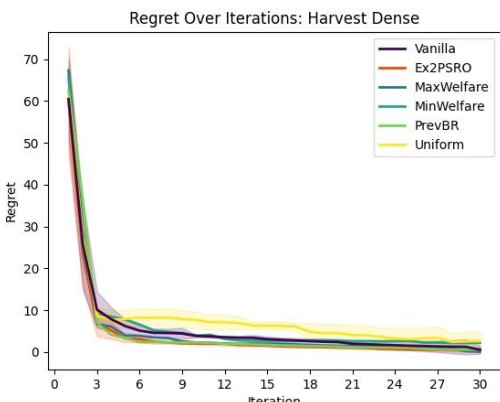

*Figure 10.* Non-truncated version of regret trends for Harvest-Sparse and Harvest-Dense.

## H.1. Regret and Welfare

We provide un-truncated versions of the plots provided in the main paper in Fig. 10. Table 8 summarizes welfare metrics in Harvest and Bargaining during hyperparameter tuning. Underlined entries correspond to metrics that are averaged over 10 trials. All other entries are averaged over 5. Bolded metrics indicate the strongest-performing configuration given $\lambda$. These tables correspond to the hyperparameter tuning mentioned in Table 5. Vanilla PSRO metrics are each averaged over 10 trials and the final metrics correspond to the best-performing vanilla PSRO parameter configuration. Ex$^2$PSRO metrics are averaged over 5 trials except for the chosen hyperparameter configuration, which is then given 5 additional trials for a total of 10. We reiterate that this parameter tuning procedure is conservative, as Ex$^2$PSRO's tuning process is susceptible to high variance and may neglect to choose a parameter setting that would have yielded even stronger performance.

*Table 8.* Welfare Tuning

| Harvest-Sparse | | | | |
|---|---|---|---|---|
| $\beta_{init}$ | $\lambda = 0$ | $\lambda = 2$ | $\lambda = 5$ | $\lambda = 10$ |
| 1e-4 | $102.36 \pm 7.53$ | $96.85 \pm 6.21$ | $108.58 \pm 6.77$ | $101.27 \pm 9.11$ |
| 3e-4 | $104.14 \pm 12.45$ | $112.39 \pm 4.11$ | $107.54 \pm 8.47$ | $118.39 \pm 5.37$ |
| 1e-3 | $\mathbf{117.81 \pm 13.90}$ | $110.35 \pm 12.64$ | $121.07 \pm 2.49$ | $113.00 \pm 10.98$ |
| 3e-3 | $116.35 \pm 4.28$ | $112.18 \pm 6.73$ | $116.94 \pm 9.19$ | $115.68 \pm 4.60$ |
| 1e-2 | $113.77 \pm 6.99$ | $\mathbf{122.18 \pm 3.42}$ | $\underline{\mathbf{129.02 \pm 7.54}}$ | $\mathbf{120.84 \pm 10.44}$ |
| Vanilla | $\underline{103.80 \pm 5.85}$ | $\underline{108.57 \pm 8.95}$ | $\underline{106.48 \pm 8.83}$ | $\underline{106.60 \pm 4.88}$ |
| **Harvest-Dense** | | | | |
| 1e-4 | $117.88 \pm 4.89$ | $117.12 \pm 2.84$ | $126.91 \pm 2.76$ | $121.52 \pm 5.57$ |
| 3e-4 | $123.05 \pm 5.74$ | $121.30 \pm 5.66$ | $125.46 \pm 8.52$ | $126.10 \pm 7.39$ |
| 1e-3 | $120.96 \pm 3.43$ | $127.27 \pm 4.06$ | $127.15 \pm 2.70$ | $125.97 \pm 3.13$ |
| 3e-3 | $134.73 \pm 4.90$ | $136.28 \pm 12.36$ | $138.82 \pm 6.03$ | $\mathbf{140.74 \pm 8.09}$ |
| 1e-2 | $\mathbf{141.46 \pm 6.30}$ | $\mathbf{137.81 \pm 4.14}$ | $\underline{\mathbf{150.22 \pm 7.86}}$ | $137.61 \pm 3.28$ |
| Vanilla | $\underline{118.98 \pm 5.67}$ | $\underline{117.38 \pm 2.30}$ | $\underline{118.36 \pm 7.81}$ | $\underline{119.57 \pm 3.35}$ |
| **Bargaining $\gamma = .99$** | | | | |
| $\beta_{init}$ | $\lambda = 0$ | $\lambda = .1$ | $\lambda = .2$ | $\lambda = .5$ |
| 1e-4 | $12.37 \pm 0.10$ | $12.23 \pm 0.58$ | $12.57 \pm 0.12$ | $12.67 \pm 0.56$ |
| 3e-4 | $12.24 \pm 0.28$ | $11.82 \pm 0.48$ | $\mathbf{12.86 \pm 0.26}$ | $12.48 \pm 0.16$ |
| 1e-3 | $12.15 \pm 0.64$ | $12.34 \pm 0.24$ | $12.75 \pm 0.15$ | $12.33 \pm 0.42$ |
| 3e-3 | $12.27 \pm 0.23$ | $12.63 \pm 0.36$ | $12.63 \pm 0.13$ | $\mathbf{12.72 \pm 0.36}$ |
| 1e-2 | $\mathbf{12.55 \pm 0.24}$ | $\mathbf{12.66 \pm 0.13}$ | $12.55 \pm 0.34$ | $12.66 \pm 0.20$ |
| Vanilla | $\underline{11.95 \pm 0.66}$ | $\underline{12.50 \pm 0.55}$ | $\underline{12.34 \pm 0.28}$ | $\underline{12.40 \pm 0.21}$ |
| **Bargaining $\gamma = .95$** | | | | |
| 1e-4 | $12.54 \pm 0.22$ | $12.83 \pm 0.42$ | $13.41 \pm 0.28$ | $13.82 \pm 0.20$ |
| 3e-4 | $13.14 \pm 0.37$ | $13.31 \pm 0.41$ | $12.28 \pm 1.08$ | $13.74 \pm 0.14$ |
| 1e-3 | $12.83 \pm 0.16$ | $13.46 \pm 0.36$ | $13.22 \pm 0.29$ | $12.74 \pm 0.71$ |
| 3e-3 | $12.51 \pm 1.05$ | $\mathbf{13.76 \pm 0.14}$ | $13.49 \pm 0.32$ | $13.32 \pm 0.29$ |
| 1e-2 | $\mathbf{13.23 \pm 0.12}$ | $12.78 \pm 0.23$ | $\mathbf{13.82 \pm 0.37}$ | $\underline{\mathbf{14.02 \pm 0.22}}$ |
| Vanilla | $\underline{12.85 \pm 0.38}$ | $\underline{12.80 \pm 0.39}$ | $\underline{13.25 \pm 0.56}$ | $\underline{13.76 \pm 0.34}$ |

## H.2. Hyperparameter Ablation

We next analyze the effect of $\beta_{init}$ and $\lambda$ on Ex$^2$PSRO's performance. We conduct ablation tests in Bargaining $\gamma = .99$ and Harvest-Sparse. We fix one of the parameters and vary the other for both environments. Bargaining $\gamma = .99$ fixes $\lambda = .2$ and $\beta_{init} = $ 3e-4 while Harvest-Sparse fixes $\lambda = 5$ and $\beta_{init} = $ 1e-2. Fig. 11 displays the regret and welfare trends over iterations, where we vary either $\lambda$ or $\beta_{init}$.

In Harvest-Sparse, results indicate that increased regularization correlates with higher-welfare equilibria. In particular, greater regularization seems to reduce how steeply welfare initially dips, a characteristic common across all trials. We posit that this arises because higher regularization encourages more coordinated abstinence from over-harvesting apples. This relationship between regularization and welfare supports that Ex$^2$PSRO, through $\beta_{init}$, has some control over the welfare of its solutions, an equilibrium selection mechanism not previously available in vanilla PSRO. In Bargaining $\gamma = .99$, relationships between regret, welfare, and tested parameters are relatively unclear. We leave it to future work to ascertain any correlations between these values.

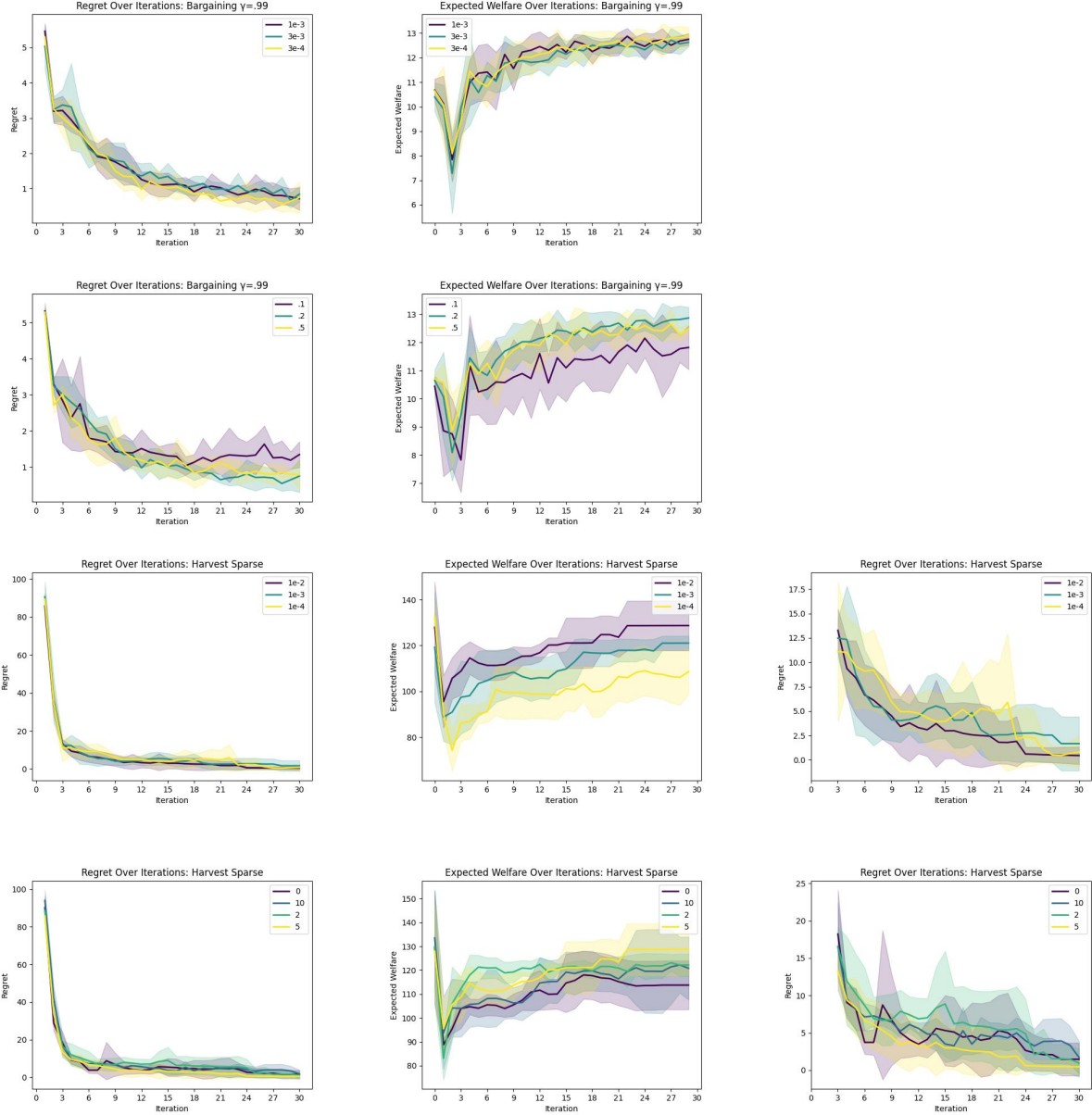

*Figure 11.* Ablation tests for Ex$^2$PSRO in Harvest Sparse and Bargaining $\gamma = .99$, varying $\lambda$ or $\beta_{init}$ and fixing the other parameter. We analyze both welfare and regret trends over iterations, displaying the data from all 30 iterations. We omit standard deviation lines for clarity.

