# OpenReview forum: "Explicit Exploration for High-Welfare Equilibria in Game-Theoretic Multiagent Reinforcement Learning"
_ICML.cc/2025/Conference — ICML 2025 poster_

### Official Review · Reviewer_RgJ4 · 2025-03-10

**Overall Recommendation:** 3

**Summary:**

This paper proposes a strategy exploration method that introduces social welfare maximization into the PSRO framework. By creating an exploration strategy and using it to regularize new strategies, E$\text{x}^2$PSRO tends to discover strategies with higher social welfare when solving the Nash equilibrium. Especially in sequential bargaining games and social dilemma games, E$\text{x}^2$PSRO demonstrates better prosocial behavior.

## update after rebuttal

Thanks for the author's reply, and my main concerns have been addressed, so I raise the point to a weak accept.

**Claims And Evidence:**

The paper provides a comprehensive explanation of the method's claims, allowing readers to understand its intentions. The experimental design is thorough and demonstrates the effectiveness of the algorithm. However, the authors have not provided an adequate analysis of the theoretical properties of the algorithm, making it difficult to determine whether the method proposed in this paper is sufficiently supported.

**Essential References Not Discussed:**

The authors have appropriately cited the latest works in this field.

**Experimental Designs Or Analyses:**

The paper provides detailed experiments that demonstrate the effectiveness of the algorithm, and I believe the experimental analysis is sufficiently thorough.

**Methods And Evaluation Criteria:**

From an application perspective, social welfare maximization is indeed an objective that can be considered when solving equilibrium strategies. The method presented in the paper may be a solution to this problem, but I believe more theoretical evidence should be provided to demonstrate that it can be effective.

**Other Comments Or Suggestions:**

My comments are given above.

**Other Strengths And Weaknesses:**

I believe the main limitation of this paper is the lack of theoretical analysis of the algorithm. For algorithms that solve equilibrium strategies in games, analyzing convergence and the results of convergence is necessary. I would like the authors to address the following questions: First, does the convergence of the E$\text{x}^2$PSRO algorithm align with that of the classical PSRO algorithm? Does introducing social welfare into the strategy exploration affect the algorithm's convergence to the Nash equilibrium? Additionally, I would like the authors to clarify whether the optimization objective in the strategy exploration presented in the paper can deterministically improve the social welfare of the equilibrium. Or, why is the method in this paper more effective in discovering equilibria with higher social welfare compared to other possible approaches (such as simultaneously using multiple meta-strategy solvers in each exploration and selecting the one with the maximum social welfare)?

**Questions For Authors:**

(1) I would like the authors to explain theoretically why the E$\text{x}^2$PSRO algorithm does not affect the convergence of the original PSRO algorithm, and whether the E$\text{x}^2$PSRO algorithm can deterministically improve social welfare when multiple different equilibria exist.

(2) I would like the authors to clarify the impact of introducing social welfare on computational complexity and whether it will consume more time in each iteration.

If there are any mistakes in my comments, please correct me. I will adjust my perspective accordingly.

**Relation To Broader Scientific Literature:**

In previous PSRO research, the primary goal was often just to solve equilibrium strategies, with little consideration given to the selection problem between different equilibria. This paper presents an algorithmic improvement addressing the potential social welfare differences between various equilibria, thus providing inspiration for research on prosocial behavior strategies.

**Theoretical Claims:**

The paper does not provide sufficient theoretical proof, which could be a limitation.

---

> ### Author Rebuttal · Authors · 2025-03-31
>
> Thank you for the thoughtful comments and feedback. As noted, we rely on experiments to validate Ex2PSRO’s faster convergence compared to classical PSRO (see Fig. 6, with more details in App. H). We do not theoretically guarantee Ex2PSRO’s performance. However, note that Ex2PSRO’s response objective differs from PSRO’s through a regularization term, and the weight on this is annealed and limited to S_{\beta} steps. Thus, Ex2PSRO naturally inherits whatever eventual convergence guarantees are provided by classical PSRO. We did not consider this a significant theoretical claim so we did not develop it explicitly.
>
> We likewise cannot guarantee deterministic social welfare improvement, and tend to doubt that is a feasible goal. As with any algorithm involving deep RL, applicable theoretical guarantees are limited and there is natural variance across trials, so equilibria found by Ex2PSRO do not have exclusively higher welfare than those of PSRO. In tested benchmarks, experiments validate that Ex2PSRO tends to improve welfare on average with significant margins relative to discovered equilibria. Just as our provided proof-of-concept MDP demonstrates a case where GRO does not improve welfare while Ex2PSRO does, there are undoubtedly environments where Ex2PSRO will not yield improvement.
>
> Ex2PSRO’s improved efficacy over alternative methods is justified in Section 4, where we motivate its design and note that high-welfare solutions may not be discovered regardless of MSS choice.
>
> For question 2, social welfare calculations are conducted during evaluation, therefore not adding computation to Ex2PSRO trials. Compared to classical PSRO in general, Ex2PSRO incurs no additional simulation expense but requires offline behavior cloning training at every iteration, which is a relatively inexpensive computation compared to online RL optimization.

---

> > ### Comment · Reviewer_RgJ4 · 2025-04-06
> >
> > Thanks for the author's reply.
> >
> > Regarding the issue of time consumption, could the authors give more specific experimental results to illustrate this, or give a computation complexity comparison theoretically. Because the explanation given in the reply seems to be not intuitive enough.
> >
> > Regarding the convergence of the equilibrium, it seems that it is not clear to show the convergence of the algorithm by the decreasing weights of the regularization part. As far as I know, the outcome of maximizing social welfare in a game is usually not a Nash equilibrium. If the algorithm eventually converges to be consistent with the original PSRO, then it seems that the regularization part would not be in effect. If the algorithm converges to some combination of high welfare strategies, how can it be guaranteed to be a stable point? I do not currently find an explanation for these questions in the text or the author's response.

---

> > > ### Author Response · Authors · 2025-04-06
> > >
> > > Thank you for the opportunity to clear up what may be a fundamental misconception. Ex2PSRO is a method for finding (approximate) equilibria with high welfare, not merely high-welfare profiles. The games of interest have multiple equilibria, which might vary quite a bit in welfare. The whole purpose of Ex2PSRO is to influence the search trajectory so that equilibria with higher welfare are more likely to be identified.  In fact, we find in our experiments (see Figs. 5 and 9) that Ex2PSRO’s final solutions have *both* higher welfare and lower regret (i.e., are closer to equilibrium) than vanilla PSRO.
> > >
> > > It is not a surprise that Ex2PSRO finds approximate equilibria, since it follows the basic structure of PSRO: interleaving empirical game-solving and approximate best-response (ABR) with deep RL. There is no direct welfare maximization, except in the selection of trajectories for creating the exploration policies. The regularization term employed for ABR just influences which of the many possible ABRs is produced in a given iteration. Our main result is that even this modest adjustment is quite effective in influencing the equilibria found, and in a positive way with respect to welfare. As noted in the original rebuttal, when Ex2PSRO has low or zero regularization strength (as it is annealed and limited), it reduces to classical PSRO. Therefore, Ex2PSRO convergence is consistent with PSRO in the sense that it will eventually converge to some equilibrium. As we claim and demonstrate, by regularizing search towards pro-welfare strategy spaces, Ex2PSRO tends to output a distribution of equilibria with higher welfare than that of PSRO.
> > >
> > > In regards to time consumption, the distinguishing factor of Ex2PSRO’s response oracle is the calculation of its regularization term (Equation 6), which incurs a cost at each gradient step. First, the training batch must also be passed through the exploration policy, which has identical complexity as querying the actor network. The divergence term computation is linear in batch size and action space size O(Ab). The actor and exploration policy network outputs consist of one node per action (size A). Therefore, the divergence approximate calculation requires not additional network passes but instead queries to the relevant, readily available output action logits. In addition, we reiterate that Ex2PSRO trains a behavior-cloned exploration policy at each iteration, which can be trained for arbitrarily long but, based on previous literature, scales linearly in the size of our dataset (trajectory buffer) and model size.

---

### Official Review · Reviewer_yU4Z · 2025-03-14

**Overall Recommendation:** 4

**Summary:**

Multi agent RL environments are getting more and more relevant. In many environments in order for solving very different tasks in the real world multiple agents are going to collaborate and solve a task. However, discovering NEs with high social welfare is hard with current RL algorithms. Even more so, discovering a diverse set of policies for these agents are hard. If we update all the models at the same time, the agents may lead to suboptimal behaviour or converging to defect defect NEs in social dilemmas. This is known since the LOLA paper which shows naive RL even fails in the very simple IPD game. Achieving a diverse set of policies is specifically hard as most of the time the diversity of the policies in a population collapses quickly. This can also lead the agents far from some high social welfare NEs as the behaviour goes extinct in the population before it gets the chance to be picked up by the RL signal. While methods like PSRO, suggest training best responses to the population and adding that best response iteratively is a nice way to achieve diversity but that does not maintain enough diversity in the population for some tasks and does not discover high social welfare policies.

This paper suggests a simple approach to this problem.  They filter out the trajectories that look promising based on a criterion, and then add another loss on the best response training loss to resemble also those filtered trajectories. It can be seen as guiding the search for best response to also explore promising training directions.

**Claims And Evidence:**

The claim is: by adding a small incentive to preserve the filtered trajectories that pass a criterion, we can incentivize better exploration. At the end, it should lead to discovery of higher social welfare NEs.

Evidence: They show results on Harvest Dense and Harvest Sparse and the bargaining game. The results are not dramatic but better than the baseline.

My concern: What is the highest possible NE in the bargaining games and the harvest games? The problem is I cannot contextualize the results by the numbers as I don't know what is the ceiling for the number and the increase itself does not look dramatic in the games. When the increases are not dramatic, minor concerns become important like whether enough seeds were used or the hyperparameters were carefully tuned.

**Essential References Not Discussed:**

I think it would be nice to mention LOLA and Best Response Shaping papers. LOLA is a fundamental paper showing naive RL fails in social dilemmas. Best Response Shaping also is another way to change the enhance best response shaping to elicit algorithms that solve social dilemmas in the sense of discovering policies with high social welfare. However, I don't want to bug the authors much because the field is vast but I thought they are important papers to notice.

**Experimental Designs Or Analyses:**

I think the design makes sense and analysis are ok. However, I think better context should be provided for the best possible NE for the environments.

**Methods And Evaluation Criteria:**

I think the evaluation makes sense. However, I wish the paper also tested their method on matrix games. However, I don't want to bug the authors with requesting experimenting on different environments as I know this mostly is frustrating.

**Other Comments Or Suggestions:**

I don't have other comments.

**Other Strengths And Weaknesses:**

I think I have mentioned everything in the above boxes. The strength is the paper is very well written and seems motivated and also the method makes sense. The weakness is I don't know if the results are dramatic or not because of the lack of what the max NE is. Also, I think hyperparameter search should have been explained in more depth. Also, I am a bit worried about the filtering mechanism. As far as I understand, different mechanisms can be chosen and the choice does really matter. I think that is a nice way to steer the exploration. However, I am not sure what happens in a mixed environment. In an environment that actually simple heuristics for flitering trajectories do not work.

**Questions For Authors:**

Q1-Can you provide a max possible return and lowest possible return in the games discussed in the paper?
Q2-If in an environment, we don't have an idea on what explorations are useful, what should we do? It seems the choice for filtering the trajectories is very important.
Q3-Is there a reason not to test the simple matrix games as most of time is done in the literature?

**Relation To Broader Scientific Literature:**

I think the paper is very relevant. multi agent RL is becoming more important and maintaining diverse policies in the population is important for effective exploration.

**Theoretical Claims:**

I have not checked the correctness but I did not observe a claim that  raised my suspicioun.

---

> ### Author Rebuttal · Authors · 2025-03-31
>
> Thank you for the insightful comments and feedback. We agree that improvements should be appropriately contextualized to determine improvement magnitude. Please refer to our answer to Q1 of reviewer uMdQ: our bar graph visual range is normalized according to an available distribution of equilibria. We don’t know the highest welfare equilibria of these games with certainty, but our experimental experience gives us reasonable evidence of the range. These bar graphs, coupled with significance tests, show that Ex2PSRO substantially outperforms vanilla PSRO considering known equilibria.
>
> We agree that exploration policy choice is vital to performance. We are not sure what the reviewer means in wondering what happens in a “mixed environment”. Our environments are all mixed in the sense of being neither completely adversarial nor common-interest.
>
> We agree with the points raised regarding additional hyperparameter search discussions and key references.
>
> Q1: Yes, Harvest generally has welfares ranging from approximately 0 and 200 while Bargaining ranges between 0 and 16.  However, as discussed above, it is more valuable to measure improvements relative to the welfare attainable in equilibrium as opposed to all possible joint policies.
> Q2: We agree that the choice of exploration policy is essential to performance. Analogous to exploration in RL, we work under the assumption that we do not know which exploration mechanisms would be effective. Otherwise, exploration would not be an issue. Our experiments demonstrate that mining experience for clues is beneficial, examine several heuristic possibilities, and validate Ex2PSRO’s MaxMin heuristic works best in practice.
> Q3: Yes, the reason is that simple matrix games are defined in terms of atomic policies and so do not provide the opportunity to exploit clues from examining constituent components of policies. The key idea of Ex2PSRO is to mine trajectory traces for policy ideas, and this could not apply in a matrix game. (Matrix games also have no use for deep RL  for strategy generation, which is a defining characteristic of vanilla PSRO.)

---

> > ### Comment · Reviewer_yU4Z · 2025-04-05
> >
> > Thanks for your answers. If I understand correctly, Ex2PSRO is tested among other mechanisms of finding an equilibria that you could come up with. Your claim is that it finds better equilibria. I think your approach is build in sense of encouraging a best response policy to be regularized to also have high rewards. I think it deserves an extra score. However, I am still a bit hesitant because I am not sure of the exact hyperparameter search setup you're doing. For example, have you done extensive hyperparameter search for Ex2PSRO while ignoring the PSRO hyperparameters? Would you mind writing out that in details?

---

> > > ### Author Response · Authors · 2025-04-05
> > >
> > > Thank you for your response. Yes, we claim that Ex2PSRO (regularization of best-response towards an exploration policy) tends to improve the welfare of discovered equilibria. The exploration policy is expressly constructed to capture high-welfare-yielding behavior observed in the normal course.
> > >
> > > You may find details regarding our hyperparameter search in Appendix A. We tuned both Ex2PSRO and the vanilla PSRO baseline, and actually made sure we gave at least as much tuning benefit to the latter. When tuning vanilla PSRO (baseline), we exhaustively tuned across lambda (RRD’s regularization term), meaning we ran 10 trials across 4 settings and immediately reported the highest welfare result. On the other hand, Ex2PSRO was tuned conservatively across lambda and Ex2PSRO’s regularization strength, where we ran 5 trials over a grid search, added 5 trials to the one initially strongest setting, and reported the result. Therefore, Ex2PSRO was at a disadvantage as a stronger Ex2PSRO setting may not have been identified but still yielded stronger results than vanilla PSRO. Ex2PSRO’s regret improvements, despite other disadvantages, are also discussed in Appendix G.
> > >
> > > The parameters for the best-response oracle shared between Ex2PSRO and PSRO are summarized in Table 2. We fix these parameters to isolate and attribute Ex2PSRO’s improvements to its introduction of regularized best-response towards an exploration policy.

---

### Official Review · Reviewer_isKQ · 2025-03-14

**Overall Recommendation:** 3

**Summary:**

The authors propose EX2PSRO, which extends PSRO to encourage finding maximum welfare solutions. A regularization target policy is trained by behavior cloning high minimum welfare trajectories collected during best-response training. This policy then acts as a regularizer during SAC best response training, encouraging similar behavior to it.

**Claims And Evidence:**

The authors claim that EX2PSRO finds higher-welfare equilibria with lower regret.

The welfare improvements are demonstrated only on two domains (Harvest and Bargaining), which limits the generality of the claims.

Additionally, many of the performance improvements are fairly subtle, which in combination with the small amount of domains, makes it difficult to tell how much of the improvements are from extensive hyperparameter tuning, etc.

**Essential References Not Discussed:**

There are no key missing references that I can discern.

**Ethical Review Concerns:**

I have no ethical concerns for this paper.

**Experimental Designs Or Analyses:**

The experimental design is sound overall, though it is disappointing how much of an effect hyperparameter tuning has on EX2PSRO and thus is required for the reported positive results.

**Methods And Evaluation Criteria:**

The EX2PSRO method proposed is intuitive and clear as an extension to PSRO. Likewise, the domains tested on are also appropriate. Welfare and regret as the primary metrics are sensible for the problem setting.

**Other Comments Or Suggestions:**

Figure 5 is confusing without axis labels. I'm assuming Regret is the x-axis, making EX2PSRO perform well compared to other methods.

**Other Strengths And Weaknesses:**

Strengths:
- The proposed approach is straightforward to implement as an extension to PSRO.
- The authors confirm an improvement to welfare in the domains tested.

Weaknesses:
- The method lacks any theoretical guarantees, and is sensitive to hyperparameters.
- Only two domains (each with two variations) are tested.

**Questions For Authors:**

To generate the exploration policy to be used as a regularizer, why not directly optimize something like $\min_p \sum^{|T|}_{t=1}r^p_t$ with reinforcement learning (offline or online)? Would this not be preferable to behavior cloning because you could then generate novel trajectories with the exploration policy that maximize welfare?

Why do you not add $\pi_{ex}^i$ to the empirical game? Would it hurt the solution?

**Relation To Broader Scientific Literature:**

This paper builds on PSRO by adding a regularizing term to try to shape the equilibrium that PSRO finds.

**Theoretical Claims:**

The theoretical justification for using behavior cloning to steer exploration is largely heuristic, and there are no formal results guaranteeing that the addition of the regularization term will lead to an improvement in welfare.

---

> ### Author Rebuttal · Authors · 2025-03-31
>
> Thank you for the helpful comments and feedback. We agree that any evaluation could benefit from additional benchmarks. However, we believe the four benchmarks (two qualitatively distinct variants on two very different games) cover a very informative range of cases. The artificial MDP example also provides insight. Ex2PSRO’s evaluation was comprehensive and prioritized thoroughness over breadth, including detail analysis of all four benchmarks with varied exploration policies as well as consideration of generalized response objectives (GRO).
>
> Regarding Ex2PSRO’s scale of improvements, we believe that Fig. 4 shows the magnitude of these in a meaningful way (see our response to Q1 of reviewer uMdQ regarding this figure). That is, Ex2PSRO generates substantial welfare improvements relative to the empirically estimated distribution of equilibria. These improvements are statistically robust, as indicated by low p-values. It is true that we employ extensive hyperparameter tuning, but, as described in Appendix H.1, we exhaustively tune the vanilla baseline but conservatively tune Ex2PSRO, meaning a stronger parameter setting may not have been identified for the latter.
>
> Regarding theoretical convergence, please refer to our response to reviewer Rgj4.
>
> Q1: The minimum of summed rewards is a non-linear objective and cannot be optimized with standard RL methods. Furthermore, online optimization would incur additional simulation expense, while offline RL algorithms are known for their relative instability, which would be exacerbated in large games.
>
> Q2: The exploration strategy pi_ex itself is usually not very good overall, even though it contains useful ingredients for an effective policy. Thus, adding it to the empirical game would not change the equilibrium, and so would not influence the trajectory of PSRO at all. We did try this in preliminary experiments and confirmed its ineffectiveness. We agree this is worth noting in the paper.

---

> > ### Comment · Reviewer_isKQ · 2025-04-09
> >
> > Thank you for your response. With the additional clarification, I am leaning more towards weak acceptance, and I have adjusted my score accordingly.

---

### Official Review · Reviewer_uMdQ · 2025-03-17

**Overall Recommendation:** 3

**Summary:**

This paper proposes the Ex2PSRO algorithm, which introduces an explicit exploration mechanism into the PSRO framework to optimize equilibrium social welfare. Specifically, the method constructs exploration policies through behavior cloning on high-welfare trajectories and guides policy optimization via KL-divergence regularization. Experimental results demonstrate that Ex2PSRO significantly improves social welfare and can be combined with existing methods to further enhance performance. This is the first equilibrium selection approach that integrates policy regularization with trajectory imitation without incurring additional simulation costs.

**Claims And Evidence:**

The central claim is well-supported: Significant welfare gains validated by statistical significance testing and ablation studies.

**Essential References Not Discussed:**

No critical omissions.

**Experimental Designs Or Analyses:**

Comprehensive experiments include baseline comparisons, ablation studies, and composition tests. Statistical rigor maintained through parameter control and repeated trials.

**Methods And Evaluation Criteria:**

Appropriate methodology leverages PSRO's iterative structure with KL regularization for directional exploration. Welfare and regret metrics effectively capture performance.

**Other Comments Or Suggestions:**

There are no more comments and suggestions.

**Other Strengths And Weaknesses:**

Strengths:
Innovative cross-method integration: First combination of policy regularization with trajectory behavior imitation, adapting behavior cloning techniques from offline RL to equilibrium selection scenarios through KL divergence.
Zero additional simulation cost: Exploration policies are entirely based on historical trajectory filtering, requiring no extra environmental interactions.
Modular design: Seamlessly combinable with existing methods like GRO through insertion of regularization terms (Table 1), demonstrating method versatility.

Weaknesses:
Symmetric game dependency: Experiments validated only on symmetric games (§2 terminology), requiring additional trajectory filtering criteria design for asymmetric scenarios, limiting generalizability.
Hyperparameter sensitivity: Requires grid search for initial regularization strength β_init and annealing steps Sβ, increasing parameter tuning difficulty in real-world deployment.

**Questions For Authors:**

The bar graph in Figure 4 normalizes welfare values to "the middle 68% of welfare found across all trials," but the rationale for this choice is unclear. Why not use absolute welfare or percentiles relative to known game equilibria?

**Relation To Broader Scientific Literature:**

Well-connected: Builds on PSRO and GRO. Policy regularization adapts offline RL to equilibrium selection. FPDE could be cited but differs in focus.

**Theoretical Claims:**

Theoretical claims are well-supported, convergence is shown via regret trends, and KL regularization aligns with imitation learning theory, validated experimentally.

---

> ### Author Rebuttal · Authors · 2025-03-31
>
> Thank you for the positive comments and feedback. We would like to clarify that Ex2PSRO is not inherently limited to symmetric games—any more than PSRO is. The methods work for non-symmetric games simply by performing strategy generation for each player separately. Assuming symmetry for this paper simplifies the notation and algorithm description, and also the presentation of experimental results (avoiding the need to show payoffs or regret for each player). The only minor twist for Ex2PSRO (as noted in Footnote 2) is that we would need to account for possible difference in utility scales across players in our criterion for trajectory selection. What we suggest is replacing raw utilities in the criterion with percentiles, measured by using the trajectory buffer as an empirical estimate of return distribution. (Interpreting welfare can also be more complicated in general for non-symmetric games, regardless of method used.)
>
> In fact, a natural description of the bargaining game is non-symmetric, between the first-moving and second-moving player. We make this symmetric ex ante by having the players flip a coin to determine who offers first. But note that our solution to this symmetric version actually contains a solution to the non-symmetric one, since our players’ policies have to cover the cases where they move first or second.
>
> Question 1: To clarify: the y-axis in Fig. 4 does show unnormalized welfare. The use of “middle 68% of welfare found across all trials” is just to decide the visual scaling (i.e, the axis range) in the plots. We elaborate our rationale in Appendix D. As noted there, we would like to calibrate this to the range of game equilibria. The actual range or distribution of these is not known for Harvest or Bargaining instances, and not apparently feasible to compute. So actually, the welfare results “found across all trials” (each in an approximate equilibrium) is our best available estimate of this equilibrium distribution.

---

### Decision · Program_Chairs · 2025-05-01

**Decision:**

Accept (poster)

**Comment:**

The authors have adequately addressed the concerns the reviewers had and are all in favor of the paper being accepted to ICML.  I concur.  The paper makes a clear contribution (finding a method to find equilibria with high welfare) to the field.